# LEARNING TO GENERALIZE ACROSS DOMAINS ON SINGLE TEST SAMPLES

**Zehao Xiao[1], Xiantong Zhen[1,2], Ling Shao[3], Cees G. M. Snoek[1]**
[1]AIM Lab, University of Amsterdam [2]Inception Institute of Artificial Intelligence
[3]National Center for Artificial Intelligence, Saudi Data and Artificial Intelligence Authority

## ABSTRACT

We strive to learn a model from a set of source domains that generalizes well to unseen target domains. The main challenge in such a domain generalization scenario is the unavailability of any target domain data during training, resulting in the learned model not being explicitly adapted to the unseen target domains. We propose learning to generalize across domains on single test samples. We leverage a meta-learning paradigm to learn our model to acquire the ability of adaptation with single samples at training time so as to further adapt itself to each single test sample at test time. We formulate the adaptation to the single test sample as a variational Bayesian inference problem, which incorporates the test sample as a conditional into the generation of model parameters. The adaptation to each test sample requires only one feed-forward computation at test time without any fine-tuning or self-supervised training on additional data from the unseen domains. Extensive ablation studies demonstrate that our model learns the ability to adapt models to each single sample by mimicking domain shifts during training. Further, our model achieves at least comparable – and often better – performance than state-of-the-art methods on multiple benchmarks for domain generalization [1].

## 1 INTRODUCTION

Despite their widespread adoption and success in academia and industry alike, deep convolutional neural networks suffer from a fundamental flaw: they have insufficient generalizability to test data that is out of their training distributions (Recht et al., 2019). Improving the generalization of machine learning methods therefore remains a challenging problem (Moreno-Torres et al., 2012; Recht et al., 2019; Krueger et al., 2021). To deal with the distribution shift, domain adaptation (Saenko et al., 2010; Long et al., 2015; Lu et al., 2020; Li et al., 2021) and domain generalization (Muandet et al., 2013; Li et al., 2017; 2020) have been extensively investigated. In domain adaptation, the key assumption is that the target data, either labeled or unlabeled, is accessible during *training*, which allows the model to be adapted to the target domains. However, this assumption does not necessarily hold in realistic scenarios where the target domain data is not available. By contrast, domain generalization strives to learn a model on source domains that can generalize well to unseen target domains without any access to the target domain data during training. We aim for domain generalization.

Previous domain generalization approaches have successfully explored domain-invariant learning (Muandet et al., 2013; Chattopadhyay et al., 2020) or domain augmentation (Shankar et al., 2018; Zhou et al., 2020a) to handle the domain shift between the source and target domains. However, since those models are trained on source domains, there will always be an "adaptivity gap" when applying them to target domains without further adaptation (Dubey et al., 2021). Thus, it is necessary to train models that are able to further adapt to target domains without ever using target data during training.

Recently, further adapting a pre-trained model using target domain data by fine-tuning or self-supervised training at test time has shown effectiveness in improving performance on domain generalization (Sun et al., 2020; Wang et al., 2021). To achieve proper adaptation to the target domain, these methods typically rely on extra fine-tuning operations on target data or extra networks (Dubey et al., 2021) at test time. Although some of these methods can achieve adaptation with a single test

---

[1] Code available: `https://github.com/zzzx1224/SingleSampleGeneralization-ICLR2022`

Table 1: **Training and test-time settings of learning methods for generalization across domains.** $(X_s, Y_s)$ and $X_t$ denote the labeled source data and unlabeled target data. $X$ indicates batches of samples, while $\mathbf{x}$ denotes just one sample. Our method does not need any target data during training, like (one-shot) domain adaptation methods, or fine-tuning operations during inference. It just uses single test samples to do per-sample adaptation for domain generalization.

| Task | Method | Training | Test-time | | | |
|------|--------|----------|-----------|---|---|---|
| | | Data | Data | Fine-tune | Extra model | Adaptive |
| Domain Adaptation | Common, e.g., (Long et al., 2015) | $X_s, Y_s, X_t$ | $\mathbf{x}_t$ | × | × | ✓ |
| | One-shot, e.g., (Luo et al., 2020) | $X_s, Y_s, \mathbf{x}_t$ | $\mathbf{x}_t$ | × | × | ✓ |
| Domain Generalization | Common, e.g., (Seo et al., 2020) | $X_s, Y_s$ | $\mathbf{x}_t$ | × | × | × |
| | Test-time training (Sun et al., 2020) | $X_s, Y_s$ | $\mathbf{x}_t$ | ✓ | ✓ | ✓ |
| | Test-time adaptation (Wang et al., 2021) | $X_s, Y_s$ | $X_t, \mathbf{x}_t$ | ✓ | × | ✓ |
| | Domain-adaptive method (Dubey et al., 2021) | $X_s, Y_s$ | $X_t, \mathbf{x}_t$ | × | ✓ | ✓ |
| | *Single sample generalization* | $X_s, Y_s$ | $\mathbf{x}_t$ | × | × | ✓ |

sample (D'Innocente et al., 2019; Sun et al., 2020; Banerjee et al., 2021), batches of target data are required for good performance. This prevents these methods from effectively adapting to the target domain when provided with very few target samples or exposed to test samples from multiple target domains without domain identifiers. Different from these works, we aim to learn the ability to adapt to a single sample such that the model can further adapt to each test sample from any target domain.

In this paper, we propose *learning to generalize on single test samples* for domain generalization. Since a single sample from the target distribution cannot inform much about the whole distribution, we consider each target sample a domain by itself. Correspondingly, we propose that each target sample adjusts the trained model in its own way. This avoids the difficulty of using the limited information available to adapt the model to the entire target domain distribution. An illustration of the method is shown in Figure 1. Using meta-learning (Li et al., 2017), we train our model to

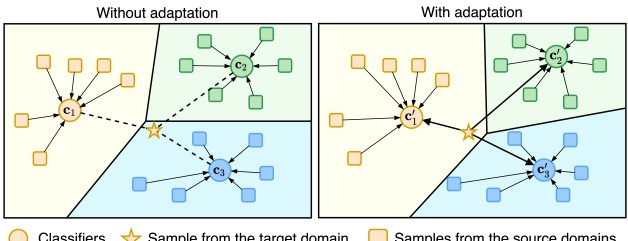

Figure 1: **Illustration of single sample generalization.** Different colors denote different classes. Circles denote the classifier prototypes for different categories. With our adaptation, the source features and the single target feature together define new classifier prototypes, which are more adapted to the target sample.

acquire the ability to adapt with each single sample on source domains such that it can then adapt to each individual sample in the target domain to handle the domain shift at test time. Unlike previous methods, our model does not need to be fine-tuned using target data or train extra networks when generalized to the unseen target domains. Once it is trained on source domains, our model is able to adapt to each single target sample with just a feed-forward computation at test time, without the need for any extra target data. Note that not a single target sample is seen during training, which does not forgo the strict source and target separation setting of domain generalization. Moreover, in contrast to previous test-time adaptation methods (Sun et al., 2020; Wang et al., 2021; Dubey et al., 2021), our method does not need to re-adapt the model to each target domain. Our model effortlessly handles test samples from multiple domains without knowing their domain identifiers.

To be specific, we build our model under the meta-learning framework and formulate the single sample generalization as a variational inference problem. In the training stage, we divide the source domains into several meta-source domains and a meta-target domain and explore the adaptive model by incorporating the information of the meta-target sample into the generation of the model parameters. For any given meta-target sample, we propose a variational distribution generated by this sample and the meta-source data to approximate the model distribution obtained by the meta-target data. In the test phase, the adapted models for the samples from the real target domains are generated on the fly by the variational model distribution. The random splits of the meta-source and meta-target domains expose the model to domain shifts and mimic the real generalization process from source domains to the target domain. Thus, the model is endowed with the ability to adapt to any unseen sample by end-to-end training with the source data only. By doing so, our method does not need to introduce any extra target data or fine-tuning operations on the target domain. A comparison of the training/test-time settings among different domain generalization (and domain adaptation) methods is summarized in Table 1.

We demonstrate the effectiveness of our single sample generalization by conducting experiments on commonly-used benchmarks for domain generalization. Our ablation studies under different domain generalization settings demonstrate the advantages of our method in comparison to other recent adaptive methods.

## 2 RELATED WORK

**Domain adaptation** To handle the domain shift between source and target domains, considerable efforts have been devoted to developing methods for domain adaptation (Long et al., 2015; Lu et al., 2020; Hoffman et al., 2018; Kumar et al., 2010; Tzeng et al., 2017; Luo et al., 2019). However, the assumption that target data is accessible during training is often invalid. Recently, some even more challenging settings were proposed in domain adaptation, e.g., few-shot (Motiian et al., 2017a) and one-shot domain adaptation (Dong & Xing, 2018; Luo et al., 2020), but there are still a few or one target sample available during training. In this work, we focus on the more strict setting, domain generalization, that assumes not a single target sample is accessible during training. We construct our model under a meta-learning framework to learn the ability of adaptation to single samples across source domains. At test time, no more learning is needed and we generate an adapted model for each individual target sample by only a forward pass, without any extra fine-tuning operation.

**Domain generalization** The problem of domain generalization was introduced by Blanchard et al. (2011), and formally posed by Muandet et al. (2013). Muandet et al. (2013) and Ghifary et al. (2016) both considered domain-invariant learning by minimizing the dissimilarity of features across source domains. Following these works, many methods explored domain invariant learning for domain generalization (Motiian et al., 2017b; Rojas-Carulla et al., 2018; Li et al., 2018b; Seo et al., 2020; Zhao et al., 2020). Alternatively, augmenting the source domain data has been explored to simulate more domain shift during training (Shankar et al., 2018; Volpi et al., 2018; Zhou et al., 2020a; Qiao et al., 2020; Zhou et al., 2020b). Our method tackles domain shift by learning to adapt to each unseen target sample, which is orthogonal to domain-invariant and domain augmentation methods.

**Domain meta-learning** Meta-learning methods have also been studied for domain generalization (Balaji et al., 2018; Du et al., 2020; Xiao et al., 2021; Du et al., 2021). Li et al. (2018a) introduced the model agnostic meta-learning (Finn et al., 2017) into domain generalization. Dou et al. (2019) further enhanced the method with global and local regularization. We follow the experimental settings of previous methods (Du et al., 2020; Xiao et al., 2021) that split source domains into meta-source and meta-target domains. By mimicking the adaptation process at training time, our model learns to learn the adaptable classifiers from meta-source domains and the meta-target sample.

**Test-time adaptation** Recently, several methods have proposed to adapt a model trained on source data to unseen target data at test time (D'Innocente et al., 2019; Sun et al., 2020; Wang et al., 2021; Dubey et al., 2021). Sun et al. (2020) trained their model on source domains with both supervised and self-supervised objective functions, and then fine-tuned it on target data with only the self-supervised one. Wang et al. (2021) further improved this method by removing the self-supervised module, while fine-tuning the model with an entropy minimization loss. Differently, Dubey et al. (2021) trained a domain embedding network with unsupervised source data, which is utilized to generate the target domain embedding by hundreds of unlabeled target samples to adapt the model at test time, without fine-tuning or further training. These adaptive methods all need extra steps, e.g., fine-tuning the model or training embedding networks, and *batches* of target samples for good adaptation. In contrast, our method learns to adapt to each target sample using just its own information without any extra target samples and fine-tuning operations. The adaptation is achieved by generating the model for each target sample with only one forward pass. Another recent work utilized test-time augmentation to improve robustness at test time (Chai et al., 2021). While it is of interest to adapt a test-time sample to source domains with such test-time augmentations, our method aims to adapt the model trained by source domains to each target sample.

## 3 METHODOLOGY

In domain generalization, several domains are defined as different data distributions in the joint space $\mathcal{X} \times \mathcal{Y}$, where $\mathcal{X}$ and $\mathcal{Y}$ denote the input space and the label space. The domains are divided into non-overlapping source domains, collectively denoted as $\mathcal{S}$ and the target domains $\mathcal{T}$. During training, only data from source domains is accessible. Data from target domains is never seen. Our goal is to learn a model on the source domains only that generalizes well to the target domains. As

we focus on homogeneous domain generalization (Zhou et al., 2021) in this work, all domains share the same label space.

Since the target data is inaccessible during training, we design a meta-learning framework with episodic training (Li et al., 2019; Balaji et al., 2018) that endows the model with the capability of learning to generalize on single test samples from any unseen domain. To this end, according to the precise domain annotations (if available) or image clusters, we divide the source domain(s) $\mathcal{S}$ into several meta-source domains $\mathcal{S}'$ and a meta-target domain $\mathcal{T}'$ in each iteration to expose the model to domain shifts. After training, the learned model is applied to the target domain $\mathcal{T}$ for evaluation. In this case, the model is exposed to different domain shifts to learn the ability to handle it. After training, the learned model is applied to the target domain $\mathcal{T}$ for evaluation.

**Variational learning** We develop our model under the probabilistic framework. Assume we have a single sample $\mathbf{x}_{t'}$ from the meta-target domain $\mathcal{T}'$, and we would like to predict its class label $\mathbf{y}_{t'}$. We consider the conditional predictive log-likelihood $\log p(\mathbf{y}_{t'}|\mathbf{x}_{t'}, \mathcal{T}')$. By incorporating the model $\boldsymbol{\theta}_{t'}$ into the prediction distribution, we have

$$\log p(\mathbf{y}_{t'}|\mathbf{x}_{t'}, \mathcal{T}') = \log \int p(\mathbf{y}_{t'}|\mathbf{x}_{t'}, \boldsymbol{\theta}_{t'})p(\boldsymbol{\theta}_{t'}|\mathcal{T}')d\boldsymbol{\theta}_{t'}, \tag{1}$$

where we condition the prediction on $\mathcal{T}'$ to leverage information from the meta-target data and $(\mathbf{x}_{t'}, \mathbf{y}_{t'})$ denotes the input-output pair of the meta-target sample. We refer to $p(\boldsymbol{\theta}_{t'}|\mathcal{T}')$ as the meta-prior distribution since it is conditioned on the meta-target data.

However, in practice, the labeled data from the target domain is inaccessible, leading to an intractable distribution $p(\boldsymbol{\theta}_t|\mathcal{T})$ during inference. To address the problem, we propose to approximate the model function $p(\boldsymbol{\theta}_t|\mathcal{T})$ with the accessible source data by the variational inference approach, which provides a framework to approximate intractable distributions via optimization. By training under the meta-learning framework, our method will learn the ability to approximate the model functions of unseen target domains by source domain data.

**Variational domain generalization** We introduce a variational distribution $q(\boldsymbol{\theta}_{t'}|\mathcal{S}')$ to approximate the true posterior over the model parameters $\boldsymbol{\theta}$. By incorporating $q(\boldsymbol{\theta}_{t'}|\mathcal{S}')$ into eq. (1), we derive a lower bound of the conditional predictive log-likelihood:

$$\log p(\mathbf{y}_{t'}|\mathbf{x}_{t'}, \mathcal{T}') = \log \int p(\mathbf{y}_{t'}|\mathbf{x}_{t'}, \boldsymbol{\theta}_{t'})p(\boldsymbol{\theta}_{t'}|\mathcal{T}')d\boldsymbol{\theta}_{t'}$$
$$\geq \mathbb{E}_{q(\boldsymbol{\theta}_{t'})}[\log p(\mathbf{y}_{t'}|\mathbf{x}_{t'}, \boldsymbol{\theta}_{t'})] - \mathbb{D}_{\mathrm{KL}}[q(\boldsymbol{\theta}_{t'}|\mathcal{S}')||p(\boldsymbol{\theta}_{t'}|\mathcal{T}')]. \tag{2}$$

The introduced Kullback-Leibler (KL) divergence term is minimized to reduce the distance between $q(\boldsymbol{\theta}_{t'}|\mathcal{S}')$ and $p(\boldsymbol{\theta}_{t'}|\mathcal{T}')$, which encourages the model $\boldsymbol{\theta}$ inferred by $q(\boldsymbol{\theta}_{t'}|\mathcal{S}')$ from meta-source domains to be generalizable to the meta-target domain.

Although eq. (2) reduces the difference between the variational distributions $q(\boldsymbol{\theta}_{t'}|\mathcal{S}')$ and $p(\boldsymbol{\theta}_{t'}|\mathcal{T}')$, $q(\boldsymbol{\theta}_{t'}|\mathcal{S}')$ still has limited capacity of handling the meta-target samples due to "adaptivity gap" (Dubey et al., 2021). Since only little information about the meta-target domain $\mathcal{T}'$ and the meta-target sample $\mathbf{x}_{t'}$ is available in $\mathcal{S}'$, there is no guarantee that the variational distribution $q(\boldsymbol{\theta}_{t'}|\mathcal{S}')$ will be adapted to the meta-target domain $\mathcal{T}'$. To reduce the "adaptivity gap", the model needs to acquire the ability to effectively use the target information. Thus, we further propose to learn to adapt the model $\boldsymbol{\theta}_{t'}$ by taking more information of the meta-target data into account.

**Single sample generalization** In domain generalization, especially for real applications, when given data from an unseen target domain, the only accessible information of the target data is from the target sample, without any annotations. We therefore propose to utilize the information contained in the given target sample to construct the adapted model. Instead of utilizing $q(\boldsymbol{\theta}_{t'}|\mathcal{S}')$ as the variational distribution, we incorporate the meta-target sample $\mathbf{x}_{t'}$ into the variational distribution $q(\boldsymbol{\theta}_{t'}|\mathbf{x}_{t'}, \mathcal{S}')$ to approximate the true posterior distribution. Given an unlabeled sample $\mathbf{x}_{t'}$ from the meta-target domain $\mathcal{T}'$, we have a new evidence lower bound (ELBO) as follows:

$$\log p(\mathbf{y}_{t'}|\mathbf{x}_{t'}, \mathcal{T}') = \log \int p(\mathbf{y}_{t'}|\mathbf{x}_{t'}, \boldsymbol{\theta}_{t'})p(\boldsymbol{\theta}_{t'}|\mathcal{T}')d\boldsymbol{\theta}_{t'}$$
$$\geq \mathbb{E}_{q(\boldsymbol{\theta}_{t'})}[\log p(\mathbf{y}_{t'}|\mathbf{x}_{t'}, \boldsymbol{\theta}_{t'})] - \mathbb{D}_{\mathrm{KL}}[q(\boldsymbol{\theta}_{t'}|\mathbf{x}_{t'}, \mathcal{S}')||p(\boldsymbol{\theta}_{t'}|\mathcal{T}')]. \tag{3}$$

By conditioning on the given sample $\mathbf{x}_{t'}$ and the source data $\mathcal{S}'$, the adapted variational distribution $q(\boldsymbol{\theta}_{t'}|\mathbf{x}_{t'}, \mathcal{S}')$ contains both the categorical information provided by the meta-source data and the

domain information given by the meta-target sample. This to a large extent guarantees the inferred model to be discriminative and adapted to the given meta-target domain sample. Moreover, the KL divergence in eq. (3) further acts as a regularizer to guide the variational distribution to be similar to the parameter distribution $p(\boldsymbol{\theta}_{t'}|\mathcal{T}')$ obtained by the meta-target data, which is assumed to be the most suitable parameter for the meta-target sample $\mathbf{x}_{t'}$. This makes the model better adapted to $\mathbf{x}_{t'}$

Based on the meta-learning framework and eq. (3), our model learns the ability to adapt the meta-source model to each meta-target instance across different domain shifts. The learned ability is exploited to handle domain shifts at test time by adapting the source model to each target instance. During inference, our method directly uses the variational distributions $q(\boldsymbol{\theta}_t|\mathbf{x}_t, \mathcal{S})$ as the adapted model generated by the information of the given target sample $\mathbf{x}_t$ and the data from all source domains $\mathcal{S}$. Thus, the method is able to generalize to each unseen target sample without the requirement of any extra data from the target domains or fine-tuning operations. The derivations of eq. (2) and eq. (3), as well as more discussions on their tightness are provided in Appendix A.

**Learning to adapt classifiers** The variational inference framework in eq. (3) indicates that we can learn to adapt all model parameters $\boldsymbol{\theta}$ to the target sample. This could, however, be computationally infeasible due to the large number of parameters in the network. Thus, for computational efficiency and feasibility we utilize the features from the feature extractor as the sample-specific information to construct the adapted classifier, which contains much fewer parameters.

To do so, we divide the model $\boldsymbol{\theta}$ into a feature extractor $\boldsymbol{\phi}$ and a classifier $\mathbf{w}$. The feature extractor is shared across domains, while the classifier is trained to be adapted to each single sample by eq. (3). Both the meta-prior distribution $p(\mathbf{w}_{t'}|\mathcal{T}')$ and the variational posterior distribution $q(\mathbf{w}_{t'}|\mathbf{x}_{t'}, \mathcal{S}')$ of the classifier $\mathbf{w}_{t'}$ are generated by amortized inference using the amortization technique (Kingma & Welling, 2013; Gershman & Goodman, 2014; Gordon et al., 2019; Shen et al., 2021). The amortization technique provides a natural way to generate model parameters by feature representations, which is one appropriate solution for our method to incorporate the target feature into the construction of the adapted variational posterior distribution. Moreover, the amortization networks can be trained and evaluated together with the model, without introducing extra training steps and fine-tuning operations to achieve the adaptation ability. Specifically, we use the center features of samples in each class from the meta-target domain and meta-source domains as the representatives of $\mathcal{T}'$ and $\mathcal{S}'$. We generate the meta-prior distribution $p(\mathbf{w}_{t'}|\mathcal{T}')$ by taking the center features of $\mathcal{T}'$ as input to the amortized inference network $\boldsymbol{\theta}_a$. For the variational distribution $q(\mathbf{w}_{t'}|\mathbf{x}_{t'}, \mathcal{S}')$, the input to the amortized inference network is the concatenation of the center features of the meta-source domain $\mathcal{S}'$ with the features of the target sample $\mathbf{x}_{t'}$ for adaptation. The amortized inference network $\boldsymbol{\theta}_a$ is shared for the inference of both the meta-prior and posterior distributions.

However, the direct concatenation operation is not necessarily optimal as the input for the amortized inference network. We propose to generate the parameter distribution using the meta-source domains and then adapt it with the given meta-target sample. To do so, we first generate a classifier distribution $p(\mathbf{w}_{s'}|\mathcal{S}')$ using the center features of the meta-source data $\mathcal{S}'$. After that, we take $p(\mathbf{w}_{s'}|\mathcal{S}')$ as a prior and combine it with the meta-target sample to estimate the variational posterior, as follows:

$$q(\mathbf{w}_{t'}|\mathbf{x}_{t'}, \mathcal{S}') = \int p(\mathbf{w}_{t'}|\mathbf{x}_{t'}, \mathbf{w}_{s'})p(\mathbf{w}_{s'}|\mathcal{S}')d\mathbf{w}_{s'}. \qquad (4)$$

This establishes a hierarchical variational inference of the posterior distribution. We show in our experiments that the hierarchical inference yields better results than the direct concatenation. In practice, we approximate the integral by:

$$q(\mathbf{w}_{t'}|\mathbf{x}_{t'}, \mathcal{S}') = \sum_{\ell=1}^{L} p(\mathbf{w}_{t'}|\mathbf{x}_{t'}, \mathbf{w}_{s'}^{(\ell)}), \qquad (5)$$

where $\mathbf{w}_{s'}^{(\ell)} \sim p(\mathbf{w}_{s'}|\mathcal{S}')$ and $L$ is the number of Monte Carlo samples.

The negative KL divergence term in eq. (3) tries to close the gap between the variational adaptive distribution $q(\mathbf{w}_{t'}|\mathbf{x}_{t'}, \mathcal{S}')$ and the meta-prior distribution $p(\mathbf{w}_{t'}|\mathcal{T}')$, aiming to make the variational distribution adapt better to the target data. However, there is no guarantee that the classifier produced by the meta-prior is discriminative enough for classification. To address this issue, we introduce an intermediate supervision based on the meta-prior distribution to make it as predictive as possible, by maximizing:

$$\mathbb{E}_{p(\mathbf{w}_{t'}|\mathcal{T}')}\big[\log p(\mathbf{y}_{t'}|\mathbf{x}_{t'}, \mathbf{w}_{t'})\big]. \qquad (6)$$

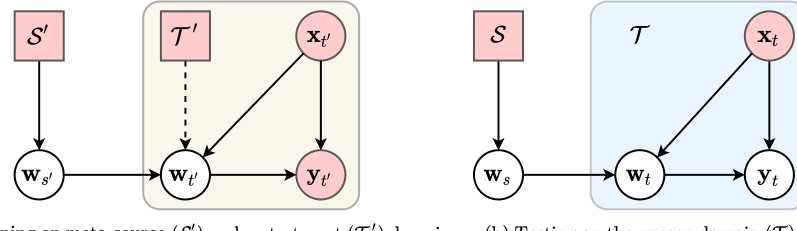

(a) Training on meta-source ($\mathcal{S}'$) and meta-target ($\mathcal{T}'$) domains    (b) Testing on the unseen domain ($\mathcal{T}$)

Figure 2: **Illustration of single sample generalization using meta-learning.** Through variational Bayesian inference, we incorporate the single sample as a conditional during the generation of model parameters. The dashed line indicates training only.

This is also implemented as a cross-entropy loss using Monte Carlo sampling from the meta-prior.

**Empirical objective function** By incorporating eq. (5) and eq. (6) into eq. (3), we obtain the final empirical objective function:

$$\mathcal{L} = \sum_{m=1}^{M} \log p(\mathbf{y}_{t'}|\mathbf{x}_{t'}, \mathbf{w}_{t'}^{(m)}) + \sum_{n=1}^{N} \log p(\mathbf{y}_{t'}|\mathbf{x}_{t'}, \mathbf{w}_{t'}^{(n)}) - \mathbb{D}_{\mathrm{KL}}(\sum_{\ell=1}^{L} p(\mathbf{w}_{t'}|\mathbf{x}_{t'}, \mathbf{w}_{s'}^{(\ell)})||p(\mathbf{w}_{t'}|\mathcal{T}'))$$

(7)

where we use Monte Carlo sampling: $\mathbf{w}_{t'}^{(m)} \sim \sum_{\ell=1}^{L} q(\mathbf{w}_{t'}|\mathbf{x}_{t'}, \mathbf{w}_{s'}^{(\ell)})$ and $\mathbf{w}_{t'}^{(n)} \sim p(\mathbf{w}_{t'}|\mathcal{T}')$.

As illustrated in Figure 2, at training time (a), we train the model episodically on meta-source domains $\mathcal{S}'$ and a meta-target domain $\mathcal{T}'$. At inference time (b), given a single sample $\mathbf{x}_t$ from the target domain $\mathcal{T}$, we use this sample to adapt the $\mathbf{w}_s$ generated by the source data. The adaptation is achieved by generating the adapted classifiers $\mathbf{w}_t$ for each target sample with only one forward pass using the learned amortization inference network $\boldsymbol{\theta}_a$. The detailed architecture is provided in Figure 3. We also provide an algorithm in Appendix B.

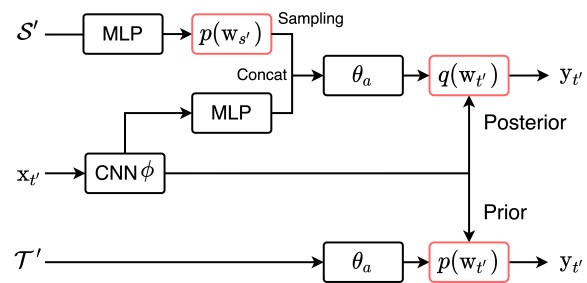

Figure 3: **Architecture of single sample generalization.** $\mathcal{S}'$ and $\mathcal{T}'$ denote meta-source and meta-target. $\mathbf{x}_{t'}$ and $\mathbf{y}_{t'}$ denote the input image and prediction of the single test sample.

## 4 EXPERIMENTS

We conduct our experiments on four widely used benchmarks for domain generalization: **PACS** (Li et al., 2017), **Office-Home** (Venkateswara et al., 2017), **rotated MNIST**, and **Fashion-MNIST** (Piratla et al., 2020). The details of the datasets and our implementations are provided in Appendix C.

**Benefit of single sample generalization** We first investigate the effectiveness of our single sample generalization on the PACS dataset. To demonstrate the benefits of all components of our method, we consider four settings: (i) the "variational amortized classifier" with eq. (2) as the objective function, (ii) our single sample generalization method without hierarchical architecture (eq. 5), (iii) single sample generalization without supervision on the meta-prior distribution (eq. 6), and (iv) the entire single sample generalization method, with the objective function in eq. (7). Results based on the ResNet-18 backbone are shown in Table 2.

By comparing rows 1 and 2 of Table 2, it is clear that our single sample generalization brings an improvement in accuracy over the invariant amortized classifier, especially on the "art-painting" and "cartoon" domains. These results demonstrate the benefits of introducing the information of the given target sample into the model generation. The comparison also demonstrates that our model learns the capability to adapt to the target sample using its own information. Comparing the last two rows shows the benefits of the intermediate supervision in eq. (6), which provides a more discriminative

Table 2: **Benefits of single sample generalization.** Ablation study on PACS using ResNet-18 averaged over five runs. Best runs within a 95% confidence margin bolded. Single sample generalization is more effective than the invariant amortized classifier. The hierarchical architecture further improves the performance. Intermediate supervision on the prior distribution is also important.

| Settings | Photo | Art-painting | Cartoon | Sketch | *Mean* |
|---|---|---|---|---|---|
| Invariant amortized classifier (eq. 2) | 95.19 ±0.30 | 79.34 ±0.65 | 78.04 ±0.85 | 76.21 ±0.54 | 82.19 ±0.35 |
| Single sample generalization (w/o eq. 5) | **95.69** ±0.20 | 81.06 ±0.33 | **79.20** ±0.49 | 77.10 ±0.64 | 83.26 ±0.31 |
| Single sample generalization (w/o eq. 6) | 95.47 ±0.20 | 80.00 ±0.61 | 77.87 ±0.91 | 77.20 ±0.88 | 82.64 ±0.32 |
| Single sample generalization (eq. 7) | **95.87** ±0.24 | **82.02** ±0.36 | **79.73** ±0.49 | **78.96** ±0.67 | **84.15** ± 0.21 |

Table 3: **Importance of appropriate domain shift during training** on PACS using ResNet-18 averaged over five runs. The more appropriate the domain shift encountered during training, the better our method will perform during inference.

| Division of source domains | Photo | Art-painting | Cartoon | Sketch | *Mean* |
|---|---|---|---|---|---|
| Split images randomly | **95.75** ±0.24 | 78.30 ±0.33 | 78.52 ±0.81 | 75.69 ±0.64 | 82.07 ±0.11 |
| Split by image clusters | 95.67 ±0.39 | 80.05 ±0.56 | **79.39** ±0.60 | 77.12 ±1.00 | 83.06 ±0.39 |
| Split by domain annotations | **95.87** ±0.24 | **82.02** ±0.36 | **79.73** ±0.49 | **78.96** ±0.67 | **84.15** ±0.21 |

Table 4: **Ablation of meta-learning setting.** The experiments are conducted on PACS using ResNet-18 averaged over five runs. Under the meta-learning framework (second row), the method performs better on all domains.

| Settings | Photo | Art-painting | Cartoon | Sketch | *Mean* |
|---|---|---|---|---|---|
| w/o Meta-learning | 94.61 ±0.65 | 79.97 ±0.43 | 78.45 ±0.29 | 75.83 ±0.79 | 82.21 ±0.36 |
| Meta-learning | **95.87** ±0.24 | **82.02** ±0.36 | **79.73** ±0.49 | **78.96** ±0.67 | **84.15** ±0.21 |

meta-prior distribution during training to learn more adapted variational distribution, especially on the "art-painting", "cartoon" and "sketch" domains. Results in rows 2 and 4 of Table 2 show that the hierarchical architecture for the variational adaptive classifier $q(\mathbf{w}_{t'})$ provides a more adapted model for each target sample and achieves the best performance on all domains.

**Importance of appropriate domain shift during training** To show the importance of mimicking an appropriate domain shift during training, we conduct experiments with different divisions of source domains during training. As shown in Table 3, when we randomly split images into source domains (row 1), the domain shift between them is too small to mimic the domain shift between source and target domains. Thus, the performance are not so good, especially on "art-painting" and "sketch". When we split the source domains with image clusters (row 2) and precise domain annotations (row 3), the performance is improved, demonstrating the importance of constructing appropriate domain shift during training. By better mimicking domain shifts during training, our method achieves better performance. Importantly, the experiments also show that our method can improve the performance without the need to have multiple domains and annotations by a simple unsupervised domain split method, e.g., clustering. The conclusion is also demonstrated in the single-source domain experiments that are provided in Table 11, Appendix D.

**Ablation of meta-learning setting** To show the importance of the meta-learning framework, we also implement a non-meta-learning version of our method. We directly generate the adapted classifiers by features from all source domains and the feature representations of the single sample. The results are shown in Table 4. Without the meta-learning framework, the model has difficulty to learn the ability to adapt to unseen domains and tends to overfit to the source domains. Thus the performance on all domains is worse than the meta-learning-based model.

**Influence of backbone** To show the influence of backbone models on the single sample generalization, we further compare our proposal with a baseline empirical risk minimization (ERM) method on PACS by varying the backbone model size. As shown in Table 5, our method achieves more obvious performance gains with larger backbone models compared to the ERM baseline. We explain this by our use of an amortized inference network that generates the classifier by the source domain features and the target sample feature. Implying an increase in backbone capacity has a direct effect on our classifier capacity, enabling better adaptation to the target sample.

**Visualization of single sample generalization** To illustrate the benefit of incorporating the test sample into classifier generation, we visualize the the target features and the classifiers of both the

Table 5: **Influence of backbone**. The experiments are conducted on PACS over five runs. Only the average accuracy of four domains is shown in the Table. With larger and larger backbone models, our method achieves better and better performance gaps compared to the ERM baseline.

| Method | AlexNet | ResNet-18 | ResNet-34 | ResNet-50 |
|---|---|---|---|---|
| ERM baseline | 71.23 $\pm$0.38 | 81.45 $\pm$0.21 | 82.58 $\pm$0.36 | 84.09 $\pm$0.13 |
| *Single sample generalization* | 72.50 $\pm$0.28 | 84.15 $\pm$0.21 | 85.91 $\pm$0.38 | 87.51 $\pm$0.22 |

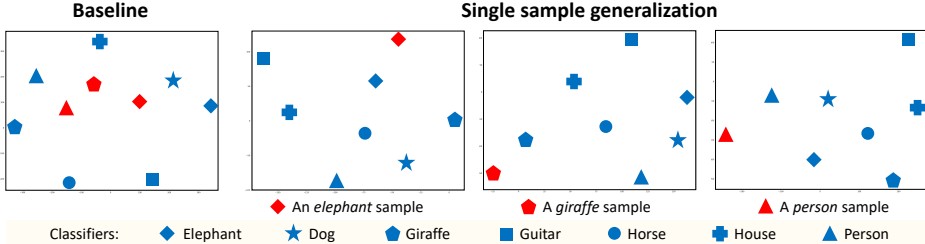

Figure 4: **Visualization of single sample generalization** on the "cartoon" domain from PACS. Different shapes denote different categories. Samples are in red, classifiers in blue. Single sample generalization generates adapted classifiers for different target samples, leading to better classification.

"variational amortized classifier" (baseline) and our single sample generalization method in Figure 4. We treat the vectors of different categories in the classifier as prototypes with the same dimension as the test features and map the classifier vectors and features into the same 2D space. As shown in the left figure (baseline), the classifier parameters of the "invariant amortized classifier" are trained on source domains and fixed for all target samples. In contrast, as shown in the three figures on the right, our method generates specific, adapted classifiers for each test target sample. For instance, the test sample (◆) has the shortest distance to the adapted classifier of its own class (◆), producing the correct predictions. More visualizations, including failure cases, are provided in Appendix E.

**Single sample generalization vs. Tent**[2] Next we conduct an experiment on rotated-MNIST to compare our method with Tent by Wang et al. (2021), which is a recent test-time adaptation method that fine-tunes the model with batches of target samples. We follow the setting in (Piratla et al., 2020) and generalize seven domains by rotating the samples by $0°$ to $90°$ in intervals of $15°$. To make the target domains more different from each other, we use images rotated by $0°$, $45°$ and $90°$ as target domains, $15°$, $30°$, $60°$ and $75°$ as source domains. For fair comparisons, we train the base model of Tent using data from all source domains, which is the same amount of train and validation data as in our method. Then, the trained base model is adapted by the target samples as in the original paper of Tent. The other settings are the same as our methods. As shown in Figure 5, we design two settings for Tent for the comparisons: "single target domain" (left) and "multiple target domains" (right). In the "single target domain" setting, we adapt the base model to each of the target domains separately using Tent and take the average accuracy of different domains. For "multiple target domains", Tent is required to adapt to all three target domains jointly without the domain identifiers. Since our method is able to adapt to each sample, the performance is consistent under both settings.

As expected, Tent achieves good adaptation for large batch sizes (e.g., 128) from a single target domain. However, when there are only very few samples, and ultimately just one sample, the adaptation of Tent starts to suffer, and may even hurt accuracy. Tent, like other test-time adaptive domain generalization methods, relies on a batch of samples to adapt to the target domain. In other words, the test samples have known domain identifiers. However, in practical settings, the test samples might come from different domains and the domain identifiers may be inaccessible. Thus, we also explore the effectiveness under the "multiple target domains" setting, as shown in Figure 5 (right). In this scenario, the adaptability of test-time adaptation methods like Tent, starts to decline, even with more samples. In contrast, our method learns the capability to adapt to each sample from any target domain, overcoming the lack of target samples while being robust under multiple target domains. We also compare our method with Tent on PACS and Office-Home. The results are shown in Table 6. We perform Tent with a baseline model trained on all source domain data, which is then fine-tuned by 128 target samples for 100 iterations. The conclusion is the same as on rotated MNIST. Our method achieves better performance. One reason can be that our model

---

[2] All Tent results are obtained by running the author-provided code accompanying (Wang et al., 2021).

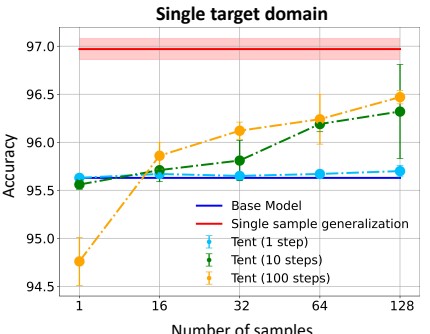 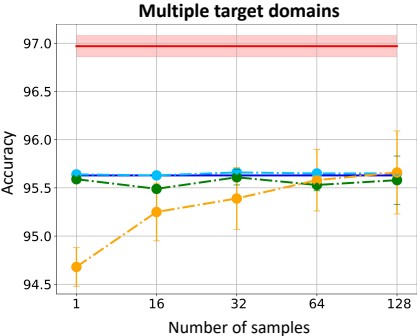

Figure 5: **Single sample generalization vs. Tent** (Wang et al., 2021) for different settings on rotated-MNIST. Tent shows good performance with a large batch of samples from a single domain (left). However, when provided only one sample or given samples from different domains without the domain id (right), Tent suffers. More detailed comparisons are provided in Appendix F.

Table 6: **Comparison on PACS and Office-Home.** Our method achieves best mean accuracy with a ResNet-50 backbone and is competitive with ResNet-18. Notably, it surpasses the adaptive methods by Wang et al. (2021) and Dubey et al. (2021), despite them using more data at test-time (Table 1).

| | PACS | | Office-Home | |
|---|---|---|---|---|
| Method | ResNet-18 | ResNet-50 | ResNet-18 | ResNet-50 |
| Carlucci et al. (2019) | 80.51 | - | 61.20 | - |
| Dou et al. (2019) | 81.04 | 82.67 | - | - |
| Zhao et al. (2020) | 81.46 | 85.34 | - | - |
| Zhou et al. (2020a) | 82.83 | 84.90 | 65.63 | 67.66 |
| Seo et al. (2020) | **85.11** | 86.64 | 62.90 | - |
| Gulrajani & Lopez-Paz (2020) | - | 85.50 | - | 66.50 |
| Dubey et al. (2021) | - | 84.50 | - | 68.90 |
| Wang et al. (2021)[†] | 83.09 ±0.13 | 86.23 ±0.22 | 64.13 ±0.16 | 67.99 ±0.22 |
| ***Single sample generalization*** | 84.15 ±0.21 | **87.51** ±0.22 | **66.02** ±0.28 | **71.07** ±0.31 |

adapts the model to each individual target sample rather than the entire target domain. Thus, different adaptations are conducted on different target samples. Alternatives are sensitive to the chosen batch of samples, which is not necessarily generalizable to the entire domain, especially when the batch is small. Another reason is our choice for meta-learning. By mimicking the adaptation process under the meta-learning framework, the model learns the ability to adapt to a single target sample. Without meta-learning, as shown in Table 4, the performance drops.

**Comparisons with state of the art** Finally, we compare our method with the state-of-the-art on four datasets. The results on PACS and Office-Home are reported in Table 6. Our method achieves good performance with both the ResNet-18 and ResNet-50 backbones. Compared with others, our method shows larger advantages with ResNet-50 than ResNet-18 on both PACS and Office-Home. This is reasonable since we generate the adaptive model using the features of source and single target samples, which contain more information with ResNet-50, leading to more adapted models. This is also demonstrated in Table 5. Results per domain, as well as the results on rotated-MNIST and Fashion-MNIST are provided in Appendix D.1, with similar results and conclusions.

## 5 CONCLUSION

We propose learning to generalize by single test samples for domain generalization. Our model is able to adapt to any unseen target sample by leveraging the information of the given sample to generate the model. We formulate the adaptation process as a variational inference problem and train the model under the meta-learning framework. After training on the source data, our method can adapt to each sample from the unseen target domains without any fine-tuning operations or extra target data. Ablation studies and further experiments on several benchmarks show the effectiveness of our method.

## ACKNOWLEDGMENT

This work is financially supported by the Inception Institute of Artificial Intelligence, the University of Amsterdam and the allowance Top consortia for Knowledge and Innovation (TKIs) from the Netherlands Ministry of Economic Affairs and Climate Policy.

## ETHICS STATEMENT

With the capability to handle the domain shift without the accessibility of target data during training and fine-tuning operations, our method has potential positive impacts in the applications that often face unseen domains in practice, e.g., medical imaging and automatic driving. Accordingly, our method would also potentially face some negative social impacts accompanying with applications, e.g., lack of fairness with the model trained by incomplete data and the privacy of patients in medical imaging.

## REPRODUCIBILITY STATEMENT

To reproduce our method, the details of the method and loss functions are provided in Section 3. We also provide the derivations and discussions about the evidence lower bound used in our method in Appendix A. The datasets used in this paper and the detailed implementations of the method are reported in Appendix C. An illustration of the architecture is shown in Figure 3. We also provide a link to the source code of our method in the Abstract.

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

# A    DERIVATIONS

**Derivation of variational domain generalization**    In domain generalization, since the target domain data is inaccessible, the distribution $p(\boldsymbol{\theta}_t|\mathcal{T})$ is intractable. We leverage the meta-learning setting to mimic the domain generalization process. In particular, we episodically divide the source domains into meta-source domains $\mathcal{S}'$ and a meta-target domain $\mathcal{T}'$ and learn the ability to generalize to the target data. Thus, during training, we generate the parameter distribution by the meta-source data $\mathcal{S}'$ to approximate the distribution $p(\boldsymbol{\theta}_{t'}|\mathcal{T}')$ generated by the meta-target data. Then at inference, the model is able to approximate the distribution $p(\boldsymbol{\theta}_t|\mathcal{T})$ with the parameters generated by the source data $\mathcal{S}$. To do so, we introduce a variational distribution $q(\boldsymbol{\theta}_{t'}|\mathcal{S}')$ to approximate the true posterior over the model parameter $\boldsymbol{\theta}$. The full derivation of eq. (2) in the main paper is as follows:

$$
\begin{aligned}
\log p(\mathbf{y}_{t'}|\mathbf{x}_{t'}, \mathcal{T}') &= \log \int p(\mathbf{y}_{t'}|\mathbf{x}_{t'}, \boldsymbol{\theta}_{t'}) p(\boldsymbol{\theta}_{t'}|\mathcal{T}') d\boldsymbol{\theta}_{t'} \\
&= \log \int p(\mathbf{y}_{t'}|\mathbf{x}_{t'}, \boldsymbol{\theta}_{t'}) \frac{p(\boldsymbol{\theta}_{t'}|\mathcal{T}')}{q(\boldsymbol{\theta}_{t'}|\mathcal{S}')} q(\boldsymbol{\theta}_{t'}|\mathcal{S}') d\boldsymbol{\theta}_{t'} \\
&= \log \left[ \mathbb{E}_{q(\boldsymbol{\theta}_{t'}|\mathcal{S}')} \left[ p(\mathbf{y}_{t'}|\mathbf{x}_{t'}, \boldsymbol{\theta}_{t'}) \frac{p(\boldsymbol{\theta}_{t'}|\mathcal{T}')}{q(\boldsymbol{\theta}_{t'}|\mathcal{S}')} \right] \right] \\
&\geq \mathbb{E}_{q(\boldsymbol{\theta}_{t'}|\mathcal{S}')} \left[ \log \left[ p(\mathbf{y}_{t'}|\mathbf{x}_{t'}, \boldsymbol{\theta}_{t'}) \frac{p(\boldsymbol{\theta}_{t'}|\mathcal{T}')}{q(\boldsymbol{\theta}_{t'}|\mathcal{S}')} \right] \right] \\
&= \mathbb{E}_{q(\boldsymbol{\theta}_{t'}|\mathcal{S}')} [\log p(\mathbf{y}_{t'}|\mathbf{x}_{t'}, \boldsymbol{\theta}_{t'})] - \mathbb{D}_{\mathrm{KL}}[q(\boldsymbol{\theta}_{t'}|\mathcal{S}')||p(\boldsymbol{\theta}_{t'}|\mathcal{T}')].
\end{aligned}
\tag{8}
$$

**Derivation of single sample generalization**    Since there is little information available about the target domain or target data in the variational distribution $q(\boldsymbol{\theta}_{t'}|\mathcal{S}')$ in eq. (2), the resultant model is not adaptive to the target domain data. To generate a model adapted to the target sample, we introduce information of the target data, and incorporate the meta-target sample $\mathbf{x}_{t'}$ into the variational distribution $q(\boldsymbol{\theta}_{t'}|\mathbf{x}_{t'}, \mathcal{S}')$. The detailed derivation is:

$$
\begin{aligned}
\log p(\mathbf{y}_{t'}|\mathbf{x}_{t'}, \mathcal{T}') &= \log \int p(\mathbf{y}_{t'}|\mathbf{x}_{t'}, \boldsymbol{\theta}_{t'}) p(\boldsymbol{\theta}_{t'}|\mathcal{T}') d\boldsymbol{\theta}_{t'} \\
&= \log \int p(\mathbf{y}_{t'}|\mathbf{x}_{t'}, \boldsymbol{\theta}_{t'}) \frac{p(\boldsymbol{\theta}_{t'}|\mathcal{T}')}{q(\boldsymbol{\theta}_{t'}|\mathbf{x}_{t'}, \mathcal{S}')} q(\boldsymbol{\theta}_{t'}|\mathbf{x}_{t'}, \mathcal{S}') d\boldsymbol{\theta}_{t'} \\
&= \log \left[ \mathbb{E}_{q(\boldsymbol{\theta}_{t'}|\mathbf{x}_{t'}, \mathcal{S}')} \left[ p(\mathbf{y}_{t'}|\mathbf{x}_{t'}, \boldsymbol{\theta}_{t'}) \frac{p(\boldsymbol{\theta}_{t'}|\mathcal{T}')}{q(\boldsymbol{\theta}_{t'}|\mathbf{x}_{t'}, \mathcal{S}')} \right] \right] \\
&\geq \mathbb{E}_{q(\boldsymbol{\theta}_{t'}|\mathbf{x}_{t'}, \mathcal{S}')} \left[ \log \left[ p(\mathbf{y}_{t'}|\mathbf{x}_{t'}, \boldsymbol{\theta}_{t'}) \frac{p(\boldsymbol{\theta}_{t'}|\mathcal{T}')}{q(\boldsymbol{\theta}_{t'}|\mathbf{x}_{t'}, \mathcal{S}')} \right] \right] \\
&= \mathbb{E}_{q(\boldsymbol{\theta}_{t'}|\mathbf{x}_{t'}, \mathcal{S}'))} [\log p(\mathbf{y}_{t'}|\mathbf{x}_{t'}, \boldsymbol{\theta}_{t'})] - \mathbb{D}_{\mathrm{KL}}[q(\boldsymbol{\theta}_{t'}|\mathbf{x}_{t'}, \mathcal{S}')||p(\boldsymbol{\theta}_{t'}|\mathcal{T}')].
\end{aligned}
\tag{9}
$$

**Comparison of the two evidence lower bounds**    The main difference between eq. (8) and eq. (9) is the variational posterior $q(\boldsymbol{\theta}_{t'})$. Compared with eq. (8), our method (eq. 9) introduces the target sample $\mathbf{x}_{t'}$ into the generation of the variational posterior, which therefore achieves more adaptive classifiers for the target sample. Moreover, the tightness of the bound is also related to the variational posterior. The approximation gap between our bound (eq. 3) and the objective log-likelihood corresponds to the KL divergence between the variational posterior $q(\boldsymbol{\theta}_{t'}|\mathbf{x}_{t'}, \mathcal{S}')$ and the true posterior $p(\boldsymbol{\theta}_{t'}|\mathbf{x}_{t'}, \mathbf{y}_{t'}, \mathcal{T}')$, as shown in the following equation:

$$
log p(\mathbf{y}_{t'}|\mathbf{x}_{t'}, \mathcal{T}') = \underbrace{\mathbb{E}_{q(\boldsymbol{\theta}_{t'})}[\log p(\mathbf{y}_{t'}|\mathbf{x}_{t'}, \boldsymbol{\theta}_{t'})] - \mathbb{D}_{\mathrm{KL}}[q(\boldsymbol{\theta}_{t'}|x_{t'}, \mathcal{S}')||p(\boldsymbol{\theta}_{t'}|\mathcal{T}')]}_{ELBO} \\
+ \underbrace{\mathbb{D}_{\mathrm{KL}}[q(\boldsymbol{\theta}_{t'}|\mathbf{x}_{t'}, \mathcal{S}')||p(\boldsymbol{\theta}_{t'}|\mathbf{x}_{t'}, \mathbf{y}_{t'}, \mathcal{T}')]}_{Approximation\ Gap}.
\tag{10}
$$

Compared with eq. (2), whose approximation gap is formulated as $\mathbb{D}_{\mathrm{KL}}[q(\boldsymbol{\theta}_{t'}|\mathcal{S}')||p(\boldsymbol{\theta}_{t'}|\mathbf{x}_{t'}, \mathbf{y}_{t'}, \mathcal{T}')]$, our method takes the given sample $\mathbf{x}_{t'}$ as one of the conditions, leading to a tighter lower bound.

## B ALGORITHM OF SINGLE SAMPLE GENERALIZATION

We describe the the detailed training and test algorithm of our single sample generalization method in Algorithm 1

---

**Algorithm 1** Single-Test-Sample Generalization

---

TRAINING TIME

**Require:** Source domains $\mathcal{S} = \{D_s\}_{s=1}^S$ each with $n$ sample pairs $(\mathbf{x}_s, \mathbf{y}_s)$

**Require:** Learning rate $\lambda$; the number of iterations $N_{iter}$.

Initialize $\Theta = \{\phi, \theta_a, \psi\}$. $\phi$: ImageNet-pretrained feature extractor; $\psi$: multiple layer module.

**for** $iter$ in $N_{iter}$ **do**

$\quad \mathcal{T}' \leftarrow$ Randomly Sample($\{D_s\}_{s=1}^S, t'$);

$\quad \mathcal{S}' \leftarrow \{D_s\}_{s=1}^S \setminus \mathcal{T}'$;

$\quad$ Sample datapoints $\{(\mathbf{x}_{t'}^{(k)}, \mathbf{y}_{t'}^{(k)})\} \sim \mathcal{T}'$.

$\quad \mu_{\mathbf{w}_{t'}^{(m)}}, \sigma_{\mathbf{w}_{t'}^{(m)}} = f_{\theta_a}(\mathcal{T}'); \mathbf{w}_{t'}^{(m)} \sim \mathcal{N}(\mu_{\mathbf{w}_{t'}^{(m)}}, \sigma_{\mathbf{w}_{t'}^{(m)}})$. // Compute meta-prior $p(\mathbf{w}_{t'}|\mathcal{T}')$ as eq.(6)

$\quad \mu_{\mathbf{w}_{s'}}, \sigma_{\mathbf{w}_{s'}} = f_{\psi}(\mathcal{S}'); \mathbf{w}_{s'} \sim \mathcal{N}(\mu_{\mathbf{w}_{s'}}, \sigma_{\mathbf{w}_{s'}})$. // Compute $p(\mathbf{w}_{s'}|\mathcal{S}')$ in eq.(4)

$\quad \mu_{\mathbf{w}_{t'}^{(n)}}, \sigma_{\mathbf{w}_{t'}^{(n)}} = f_{\theta_a}(\mathbf{w}_{s'}, \phi(\mathbf{x}_{t'}^{(k)})); \mathbf{w}_{t'}^{(n)} \sim \mathcal{N}(\mu_{\mathbf{w}_{t'}^{(n)}}, \sigma_{\mathbf{w}_{t'}^{(n)}})$. // Compute $q(\mathbf{w}_{t'}|\mathbf{x}_{t'}^{(k)}, \mathcal{S}')$ as eq.(5)

$\quad \mathcal{L} = \sum\limits_{k} \Big( \sum\limits_{m=1}^M \text{CrossEntropy}\,((\mathbf{w}_{t'}^{(m)} \cdot \phi(\mathbf{x}_{t'}^{(k)})), \mathbf{y}_{t'}^{(k)}) + \sum\limits_{n=1}^N \text{CrossEntropy}\,((\mathbf{w}_{t'}^{(n)} \cdot \mathbf{x}_{t'}^{(k)}), \mathbf{y}_{t'}^{(k)})$

$\quad\quad + \mathbb{D}_{\text{KL}}[q(\mathbf{w}_{t'}|\mathbf{x}_{t'}^{(k)}, \mathcal{S}')||p(\mathbf{w}_{t'}|\mathcal{T}')] \Big)$. // Compute loss as eq.(7)

$\quad$ Update parameters: $\Theta \leftarrow \Theta - \lambda \nabla_\Theta \mathcal{L}$. // Gradient descent step

**end for**

---

TEST TIME

**Require**: Source domains $\mathcal{S} = \{D_s\}_{s=1}^S$; target samples $\{\mathbf{x}_t\}_{t=1}^T$ from the target domain $\mathcal{T}$; trained model with weight $\Theta = \{\phi, \theta_a, \psi\}$.

$\mu_{\mathbf{w}_s}, \sigma_{\mathbf{w}_s} = f_{\psi}(\mathcal{S}); \mathbf{w}_s \sim \mathcal{N}(\mu_{\mathbf{w}_s}, \sigma_{\mathbf{w}_s})$. // Compute $p(\mathbf{w}_s|\mathcal{S})$

$\mu_{\mathbf{w}_t^{(n)}}, \sigma_{\mathbf{w}_t^{(n)}} = f_{\theta_a}(\mathbf{w}_s, \phi(\mathbf{x}_t)); \mathbf{w}_t^{(n)} \sim \mathcal{N}(\mu_{\mathbf{w}_t^{(n)}}, \sigma_{\mathbf{w}_t^{(n)}})$. // Compute $q(\mathbf{w}_t|\mathbf{x}_t^{(k)}, \mathcal{S})$

**return** $\mathbf{y}_t = \mathbf{w}_t^{(n)} \cdot \mathbf{x}_t$.

---

## C DATASETS AND IMPLEMENTATION DETAILS

**Datasets** PACS (Li et al., 2017) is a widely used dataset consisting of 9,991 images of seven classes from four domains, i.e., *photo*, *art-painting*, *cartoon* and *sketch*. We use the same training and validation split as in (Li et al., 2017) and follow the "leave-one-out" protocol from (Li et al., 2017; 2019; Carlucci et al., 2019). **Office-Home** (Venkateswara et al., 2017) has 15,500 images of 65 categories. The images are also from four domains, i.e., *art*, *clipart*, *product* and *real-world*. We use the same experimental protocol as for PACS. **Rotated MNIST and Fashion-MNIST** are utilized in Piratla et al. (2020). We follow their recommended settings and randomly select a subset of 2,000 images from MNIST and 10,000 images from Fashion-MNIST. The subset of images is rotated by $0°$ through $90°$ in intervals of $15°$, creating seven domains. We use the subsets with rotation angles from $15°$ to $75°$ as the source domains, and images rotated by $0°$ and $90°$ as the target domains. For comparison, we evaluate our method on both in-distribution and out-of-distribution data.

**Implementation details** We evaluate our method on PACS with both ResNet-18 and ResNet-50 (He et al., 2016) pretrained on ImageNet (Deng et al., 2009) as the backbone. During training, we use Adam optimization and train the model for 10,000 iterations. The learning rate of the backbone is set to 0.00005 for ResNet-18 and 0.00001 for ResNet-50, while the learning rate of the network for generating the classifier is set to 0.0001 consistently. The batch size is 128. To generate the classifier, in each iteration we select 10 samples from each category in each meta-source domain for generating the center features $\mathcal{S}'$, and the same number of samples from the meta-target domain for $\mathcal{T}'$. The model with the highest validation accuracy is utilized for evaluation on the target domain. Most of the experimental settings and hyperparameters on Office-Home are the same as on PACS. Since there are more categories in Office-Home, the number of samples per category per domain is set to 5 to fit the memory footprint. The learning rate of the backbone is set to 0.00001 for both ResNet-18

Table 7: **Implementation details of our method per dataset and backbone.** "Number of source samples" denotes the number of samples per class per source domain for generating the adapted classifier.

| Dataset | Backbone | Classifier learning rate | Backbone learning rate | Number of source samples |
|---|---|---|---|---|
| PACS | ResNet-18 | 0.0001 | 0.00005 | 10 |
| | ResNet-50 | 0.0001 | 0.00001 | 10 |
| Office-Home | ResNet-18 | 0.0001 | 0.00001 | 5 |
| | ResNet-50 | 0.0001 | 0.00001 | 5 |
| Rotated MNIST | ResNet-18 | 0.0001 | 0.00005 | 10 |
| Fashion-MNIST | ResNet-18 | 0.0001 | 0.00005 | 10 |

Table 8: **Comparison on PACS.** Our method achieves best mean accuracy with a ResNet-50 backbone and is competitive with ResNet-18. Notably, it surpasses the adaptive test-time methods by Wang et al. (2021) and Dubey et al. (2021), despite them using more data at test-time (see Table 1). † denotes the reimplemented results in both this table and Table 9.

| Backbone | Method | Photo | Art-painting | Cartoon | Sketch | *Mean* |
|---|---|---|---|---|---|---|
| ResNet-18 | Carlucci et al. (2019) | 96.03 | 79.42 | 75.25 | 71.35 | 80.51 |
| | Dou et al. (2019) | 94.99 | 80.29 | 77.17 | 71.69 | 81.04 |
| | Zhao et al. (2020) | **96.65** | 80.70 | 76.40 | 71.77 | 81.46 |
| | Zhou et al. (2020a) | 96.20 | 83.30 | 78.20 | 73.60 | 82.83 |
| | Wang et al. (2021)† | 95.49 $_{\pm0.27}$ | 81.55 $_{\pm0.35}$ | 77.67 $_{\pm0.39}$ | 77.64 $_{\pm0.27}$ | 83.09 $_{\pm0.13}$ |
| | Zhou et al. (2020b) | 96.10 | 84.10 | 78.80 | 75.90 | 83.70 |
| | Seo et al. (2020) | 95.87 | **84.67** | 77.65 | **82.23** | **85.11** |
| | *This paper* | 95.87 $_{\pm0.24}$ | 82.02 $_{\pm0.36}$ | **79.73** $_{\pm0.49}$ | 78.96 $_{\pm0.67}$ | 84.15 $_{\pm0.21}$ |
| ResNet-50 | Dou et al. (2019) | 95.01 | 82.89 | 80.49 | 72.29 | 82.67 |
| | Dubey et al. (2021) | - | - | - | - | 84.50 |
| | Zhou et al. (2020a)† | 97.55 | 85.21 | 80.33 | 76.53 | 84.90 |
| | Zhao et al. (2020) | **98.25** | 87.51 | 79.31 | 76.30 | 85.34 |
| | Gulrajani & Lopez-Paz (2020) | 97.20 | 84.70 | 80.80 | 79.30 | 85.50 |
| | Wang et al. (2021)† | 97.96 $_{\pm0.27}$ | 86.30 $_{\pm0.26}$ | 82.53 $_{\pm0.69}$ | 78.11 $_{\pm0.73}$ | 86.23 $_{\pm0.22}$ |
| | Seo et al. (2020) | 95.99 | 87.04 | 80.62 | **82.90** | 86.64 |
| | *This paper* | 97.88 $_{\pm0.15}$ | **88.09** $_{\pm0.26}$ | **83.83** $_{\pm0.68}$ | 80.21 $_{\pm0.66}$ | **87.51** $_{\pm0.22}$ |

and ResNet-50. For fair comparisons, we evaluate the rotated MNIST and Fashion-MNIST with ResNet-18, following (Piratla et al., 2020). The other experimental settings are also the same as PACS. We train all models on an NVIDIA Tesla V100 GPU. The detailed information about other hyperparameters are summarized in Table 7.

# D    EXTRA EXPERIMENTS

## D.1    DETAILED COMPARISONS WITH STATE OF THE ART

**PACS** In Table 8, we conduct experiments with both ResNet-18 and ResNet-50 backbones. Our method achieves good performance using the ResNet-18 backbone. For each individual domain, we are competitive with the state of the art and slightly better on the "cartoon" domain when using ResNet-18. On the "art-painting" and "cartoon" domains with ResNet-50, our method also achieves good performance. Although it delivers competitive, or sometimes even better, performance on most domains, our method is not as good on the "sketch" domain. One reason might be that the images and features from this domain carry less information than other domains, leading to less adaptive models.

**Office-Home** As shown in Table 9, our method again achieves good overall performance with both the ResNet-18 and ResNet-50 backbones. With ResNet-18 as the backbone, we outperform other methods by a good margin on the "clipart" domain, while delivering competitive performance on the other domains. When utilizing ResNet-50 as the backbone, our method achieves good performance on all four domains.

**Rotated MNIST and Fashion-MNIST.** On rotated MNIST and Fashion-MNIST, for fair comparison, we use ResNet-18 as the backbone following (Piratla et al., 2020) and evaluate the method on both in-distribution and out-of-distribution sets. Our method achieves best performance on both datasets as

Table 9: **Comparison on Office-Home.** Our method achieves the best mean accuracy using both a ResNet-18 and ResNet-50 backbone. Notably, it surpasses the adaptive test-time methods by Wang et al. (2021) and Dubey et al. (2021), despite them using more data at test-time (see Table 1).

| Backbone | Method | Art | Clipart | Product | Real World | *Mean* |
|---|---|---|---|---|---|---|
| ResNet-18 | Carlucci et al. (2019) | 53.04 | 47.51 | 71.47 | 72.79 | 61.20 |
| | Seo et al. (2020) | 59.37 | 45.70 | 71.84 | 74.68 | 62.90 |
| | Huang et al. (2020) | 58.42 | 47.90 | 71.63 | 74.54 | 63.12 |
| | Wang et al. (2021)$^\dagger$ | 56.45 $_{\pm0.30}$ | 52.06 $_{\pm0.18}$ | 73.19 $_{\pm0.42}$ | 74.82 $_{\pm0.27}$ | 64.13 $_{\pm0.16}$ |
| | Zhou et al. (2020a) | **60.60** | 50.10 | **74.80** | **77.00** | 65.63 |
| | *This paper* | 59.39 $_{\pm0.43}$ | **53.94** $_{\pm0.45}$ | 74.68 $_{\pm0.33}$ | 76.07 $_{\pm0.17}$ | **66.02** $_{\pm0.28}$ |
| ResNet-50 | Gulrajani & Lopez-Paz (2020) | 61.30 | 52.40 | 75.80 | 76.60 | 66.50 |
| | Zhou et al. (2020a)$^\dagger$ | 63.62 | 51.48 | 76.57 | 78.95 | 67.66 |
| | Wang et al. (2021)$^\dagger$ | 62.12 $_{\pm0.32}$ | 56.65 $_{\pm0.49}$ | 75.61 $_{\pm0.57}$ | 77.58 $_{\pm0.42}$ | 67.99 $_{\pm0.22}$ |
| | Dubey et al. (2021) | - | - | - | - | 68.90 |
| | Sun & Saenko (2016) | 65.30 | 54.40 | 76.50 | 78.40 | 68.70 |
| | *This paper* | **67.21** $_{\pm0.59}$ | **57.97** $_{\pm0.37}$ | **78.61** $_{\pm0.59}$ | **80.47** $_{\pm0.16}$ | **71.07** $_{\pm0.31}$ |

Table 10: **Comparison on rotated MNIST and Fashion-MNIST.** In-distribution performance is evaluated on the test sets of MNIST and Fashion-MNIST with rotation angles of $15°$, $30°$, $45°$, $60°$ and $75°$, while the out-of-distribution performance is evaluated on test sets with angles of $0°$ and $90°$. We achieve the best performance on both the in-distribution and out-of-distribution test sets.

| | MNIST | | Fashion-MNIST | |
|---|---|---|---|---|
| | In-distribution | Out-of-distribution | In-distribution | Out-of-distribution |
| Dou et al. (2019) | 98.2 | 93.2 | 86.9 | 72.4 |
| Piratla et al. (2020) | 98.4 | 94.7 | 89.7 | 78.0 |
| *This paper* | **98.9** $_{\pm0.1}$ | **95.8** $_{\pm0.1}$ | **90.9** $_{\pm0.2}$ | **80.8** $_{\pm0.5}$ |

shown in Table 10, especially for the out-of-distribution setting. Moreover, our method demonstrates less performance drop from in-distribution to out-of-distribution data in comparison to other methods.

## D.2 SINGLE SOURCE DOMAIN GENERALIZATION

As our method learns the adaptation ability by mimicking the domain shift during training, it requires at least two source domains to support the meta-learning scheme, which is a limitation of the method.

We conduct an experiment that uses SVHN as the single source domain and MNIST as the target. We generate source domains by two simple methods: random image split and clustering of images (K-means). The results in Table 11 show that our method performs slightly better with larger domain shift simulated in the single source domain. To learn better adaptation ability, auxiliary methods are needed to generate larger domain shifts, for example by domain augmentation techniques like Qiao et al. (2020).

## E VISUALIZATIONS

**More comparisons with baseline**    To further demonstrate the effectiveness of our method, we provide more visualizations of the classifier parameters and feature representations from different domains in Figure. 6. The left column shows the visualizations of the "variational amortized classifier" (baseline), and the three columns on the right show our results. The same conclusion can be drawn here as in the main paper. Our method is able to generate adapted classifiers for each test sample, providing better predictions than the fixed classifiers in the baseline method.

**Failure cases**    To gain insights into our method, we also provide some failure cases in Figure 7. Our method gets confused when samples have objects of different categories in the same image, as shown in the first row. Although failing in these cases, our model provides correct predictions for the other objects in the image. In the visualizations, the classifier with the same category of the labeled object is also close to the feature, which demonstrates the effectiveness of our method. In the middle two rows, our method struggles with the samples that contain multiple objects or complex backgrounds.

Table 11: Experiments on single source domain generalization. We use SVHN as the source domain and MNIST as the target. Our method performs slightly better with larger domain shift simulated in the single source domain.

| Method | SVHN → MNIST |
|---|---|
| Baseline | 82.05 ±0.34 |
| Random split by images | **82.45** ±0.70 |
| Split by cluster | **83.01** ±0.36 |

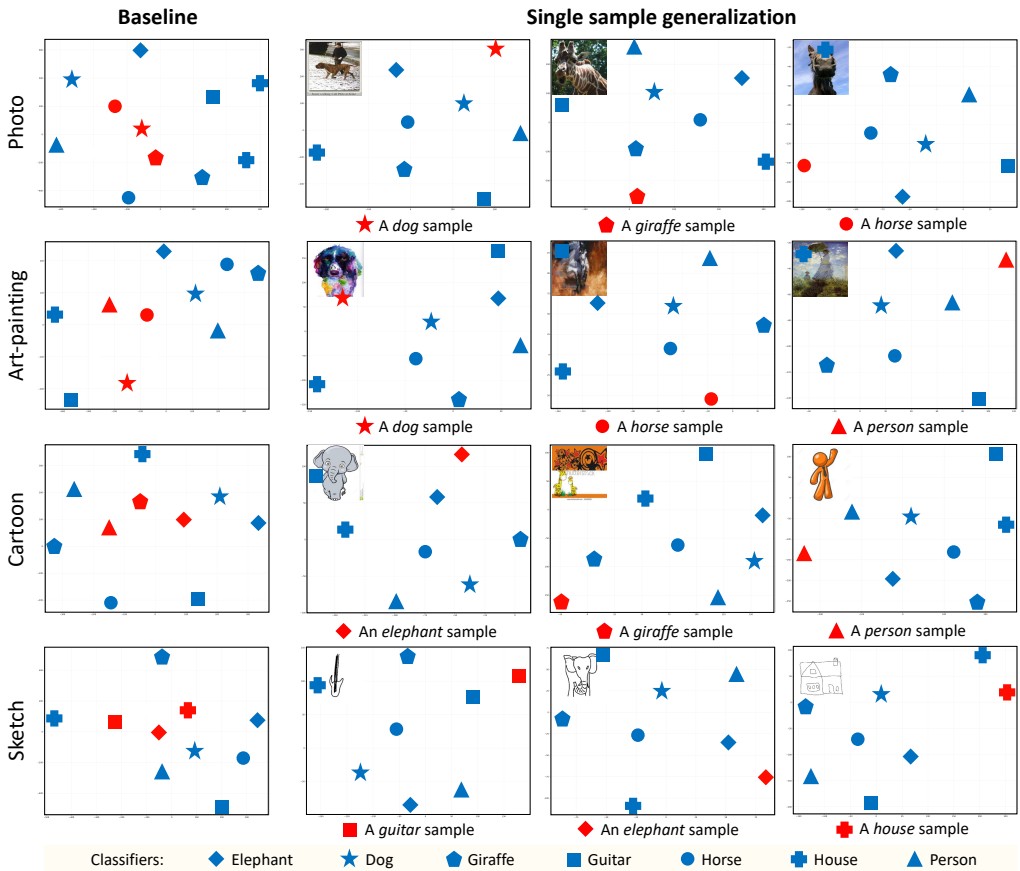

Figure 6: **Visualization of single sample generalization on PACS.** We plot the test samples and classifiers in the same 2D plane. Different shapes denote different categories. Samples are in red, classifiers in blue. The baseline method produces the same classifier for all samples from each domain, while our method generates the classifier adapted to each target sample. The two methods use the same test samples from each domain. The test sample is shown in the left-top corner of each sub-figure. The visualization shows why our model performs better.

A possible reason is that to take full advantage of the given sample, we utilize multi-level features of the target sample to generate the adapted classifier. However, the complex background and multiple objects in the image bring too much noisy information, leading to less adapted classifiers for the given sample. A solution can be extracting information of the target sample selectively to reduce noise in the features. Moreover, in the "sketch" domain, as shown in the last row, the samples usually have less information than other domains. This makes our method more sensitive to the given information. When there is noisy information, e.g., the chair in the left figure, or object with unobvious features, e.g., the elephant in the right figure, the method fails to generate proper classifiers for the sample.

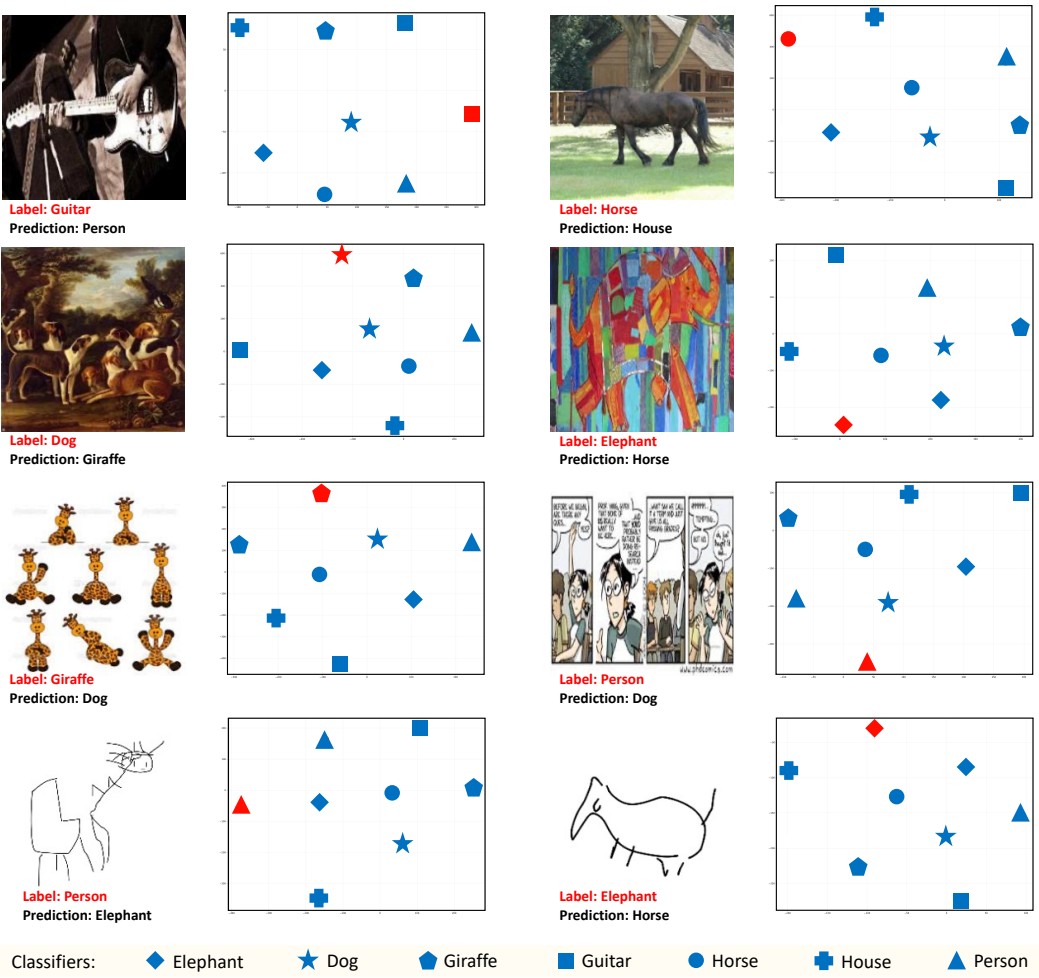

Figure 7: **Failure case visualizations of our method.** The visualization settings are the same as Figure 6. Our method makes wrong predictions, be it that the correct classifiers are also close to the samples, which shows that the classifiers generated by our method are still adapted to the given sample in these cases.

## F    MORE COMPARISONS WITH TENT

We provide more detailed comparisons with Tent Wang et al. (2021) in Figure 8 to further show the effectiveness of our method. The three rows in Figure 8 show the accuracy of different settings on 0°, 45°, and 90° target domain, respectively. The mean accuracy is shown in Figure 5 in the main paper. The same as in the main paper, the left column shows results under the "single target domain" setting, while the right column shows the "multiple target domains" setting. In the "single target domain" setting, Tent is utilized to adapt the base model trained on source domains to each target domain independently. By contrast, in the "multiple target domains" setting, the base model is adapted and evaluated on three target domains jointly, without the domain identifiers. The base model is adapted by Tent using different number of samples, e.g., 1, 32, 128, with different optimization steps, e.g., 1, 10, 100. The same conclusion can be drawn here as in the main paper. Compared with Tent, our method adapts the model to each target sample individually by a single and efficient feed-forward pass. Thus, our performance is independent to the settings and optimization steps.

Specifically, as shown in the left column, Tent achieves better performance with both larger numbers of samples and fine-tuning steps, especially on 0° and 90° domains. However, with fewer target samples being available, and ultimately just one sample, the adaptation of Tent starts to suffer, or even hurt the accuracy with more optimization steps. When adapted to three target domains jointly, as shown in the right column, the adapted performance is similar with the "single target domain"

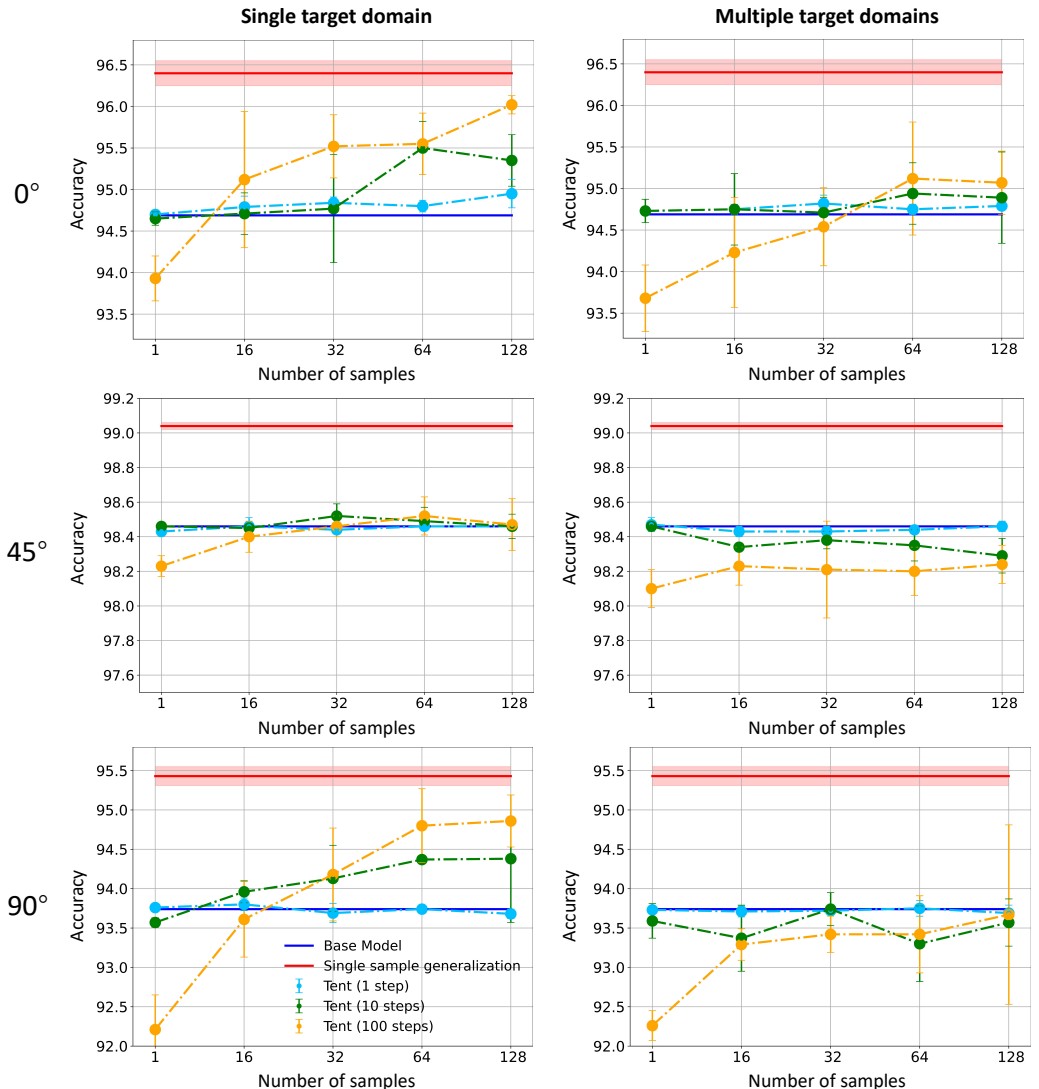

Figure 8: **Single sample generalization vs. Tent for different settings on rotated-MNIST.** Tent shows good performance with a large batch of samples from a single domain (left). When provided only one sample or when given samples from different domains, without their domain id (right), Tent suffers. In contrast, our method is independent to the number of target samples and domains.

settings with few target samples. With more target samples, the performance degrades, worse than the left column. Accuracy on $0°$ domain improves slightly, while on $45°$ and $90°$ domain even drops as the adaptation is affected by samples from other domains. In contrast, since our method adapts the model to each target sample individually, we achieve consistently good performance regardless of how many domains the test samples cover. This demonstrates the benefit of our single sample generalization.

