# OpenReview forum: "Learning to Generalize across Domains on Single Test Samples"
_ICLR.cc/2022/Conference — ICLR 2022 Poster_

### Official Review · Reviewer_xA1m · 2021-10-22

**Correctness:** 3
**Technical Novelty And Significance:** 3
**Empirical Novelty And Significance:** 3
**Recommendation:** 5
**Confidence:** 3

**Main Review:**

- (+) Interesting setting that has not received much attention: test-time adaptation from a *single* test example.
- (-) However, there is little intuition given why test-time adaptation using a *single* example makes sense. It's not clear why this should work, unlike adaptation using a batch of examples, which makes much more sense intuitively (to me, at least).
- (+) Elegant formulation of the method as variational Bayesian inference.
- (+) Computationally-efficient method: a single forward pass necessary to adapt the model
- (-) I'm not sure the method is really an instance of "meta-learning" (as in "learning to learn"). I understand that the method is simply mimicking the adaptation process at training time so as to optimize the parameters of the adaptation process.
- (-) There is little analysis of the results to show why the method performs quantitatively so well. It's difficult for me to intuitively grasp why adaptation using a single test example outperforms methods that also use other test examples from the same domain (Wang et al. 2021, Dubey et al. 2021).

Additional advice to improve the paper.

- Illustrating figures (as early as the introduction) would be helpful to convey more efficiently the gist of the method, and the motivation for the setting (as mentioned above, it's not clear to me why adaptation to a single example makes sense).
- Additional possible reference of a method for test-time adaptation: [Learning to Generalize One Sample at a Time with Self-Supervision](https://arxiv.org/abs/1910.03915)

**Summary Of The Paper:**

- The paper describes a method for domain generalization that performs test-time adaptation using a single test example at a time (as opposed to a transductive setting used in other works where a whole batch of test examples are used).
- The method is cast as a meta learning task. The training data is split into meta-training/test domains to mimic the adaptation process during training.

**Summary Of The Review:**

Preliminary recommendation: unsure. The paper describes an elegant method with good empirical performance, but it does not provide intuition/theoretical explanations/empirical analysis why it should/does work. I feel that this paper brings little new scientific knowledge that future research could build on, hence its impact may be limited.

Requests/questions to the authors.
- Please address the negative points raised above.
- How are the source/target domains generated during training ? I understand they are generated at random, but since you have multiple domains in the datasets used (say, PACS) I imagine you do use the annotations of these domains at some point ?

---
**Final summary after discussion with other reviewers**

I am fine seeing this paper accepted given that this is the advice of the other reviewers. However I still see important weaknesses in this paper that the authors may want to fix in the final version and/or in future work.

1. There is no intuitive or strong theoretical support for adaptation from a single test example. The proposed method does produce non-trivial empirical results that means that it must rely on specific information/assumptions, which are not made clear. The reason it matters is that these assumptions must have limits of applicability. These **limitations** should be explored, discussed, or at the very least acknowledged in the paper. My impression is that the variational and meta learning aspects of the proposed implementation are eclipsing the more fundamental points on which the method relies.


2. The authors responded to the lack of **illustrations** by including fig. 1/2 but I don't find these satisfatory. Figure 1 provides almost no information regarding (1) above. As for Figure 2, it's not even clear what it represents ("architecture of our method"). Training time ? Test time ? The boxes are a mix of operations/layers, variables, and distributions. The colors are not defined. Etc.


3. I still think that calling of the approach "**meta learning**" is a stretch. Another reviewer made me realize that there is a precedent in the literature for using the term when merely mimicking the adaptation at training time, but I'm not sure that perpetuating a bad choice is a valid reason.

---

> ### Author Response · Authors · 2021-11-22
> **Response to Reviewer xA1m**
>
>
> **Intuition of our method**
>
> Our intuition is that a single target sample provides target information that can be used to achieve model adaptation to the sample itself. Hence, our method adapts the model to each individual target sample, rather than the entire target domain. Further, our meta-learning scheme enables the model to learn the ability to handle domain shifts by single samples. We added the intuition in the Introduction.
>
> **Meta-learning scheme of the method**
>
> We would like to clarify that our method follows the spirit of “learning to learn”. By mimicking the adaptation process at training time, the model learns to learn the adaptable classifiers from meta-source domains and the meta-target sample (as mimicked by another source domain during training). We follow the experimental settings of previous methods that split source domains into meta-source and meta-target domains (e.g., Li et al. AAAI 2018, Dou et al. NeurIPS 2019).  We added these clarifications in related work.
>
> **Analysis of the results compared with other adaptation methods**
>
> Our method performs well because it adapts the model to each individual target sample rather than the entire target domain (or a batch). Thus, different adaptations are conducted on different target samples. Adaptation with a batch is sensitive to the chosen batch of samples, which is not necessarily generalizable to the entire domain, especially when the batch is small. Another reason is our choice for meta-learning. By mimicking the adaptation process under the meta-learning framework, the model learns the ability to adapt to a single target sample. Without meta-learning, as hinted by Reviewer 6n8V and shown in Table 6 in Appendix B, the performance drops.
> We added these analyses in Section 4.
>
> **Generation of the source and target domains**
>
> We apologize that the generation of source and target domains are not described clearly. Indeed, we use the domain annotations to split the source data by domain into meta-source and meta-target domains and obtain a separate unseen domain as the target domain. See also our response to similar comments by Reviewer n6z3 and Reviewer wnp4. We clarified the generation of source and target domains in Section 3.
>
> **Additional advice**
>
> We added an illustration of our method in Figure 1 and cited the reference in Introduction and Related Work. Thank you.

---

> > ### Comment · Reviewer_xA1m · 2021-11-22
> > **Elements still not clear**
> >
> > Thanks for the response and updated draft.
> >
> > > We added the intuition in the Introduction.
> >
> > Thanks but the response provided, as well as the addition to the introduction are simply restating what was already in the abstract/paper. I still cannot find an intuitive justification for the approach.
> >
> > At a fundamental level (independent from the meta learning, Bayesian inference, and other specifics of this paper), adaptation to a distribution shift requires some knowledge of the relation between the source and target distributions (i.e. what can be transferred). Obviously a *single sample* from the target distribution can't inform much about the whole target distribution, and even less so about its relation with the source distribution. The only way I can see the approach work is by narrowly restricting the distribution shifts it could handle. Could this be what the meta learning part is doing ? In this case, can you characterize the limits of the method (i.e. the distribution shifts that it can handle, given those seen in meta training) ? Or am I just mistaken in my understanding ?
> >
> >
> > > the model learns to learn the adaptable classifiers from meta-source domains and the meta-target sample (as mimicked by another source domain during training).
> >
> > I maintain my objection that it's a stretch to call this meta learning. I can demonstrate this by removing the "meta"-related words from the sentence, and it still makes perfect without the meta learning jargon.
> >
> > "the model learns to adapt classifiers from source domains and the target sample (as mimicked by another source domain during training)"
> >
> >
> > > Generation of the source and target domains
> >
> > Let me just double-check my understanding, for example with PACS.
> > - For the evaluation on "paintings", you train the model on Photo/Cartoon/Sketch, using alternatively any 2 of these 3 as source (e.g. P/C) and a target sample from the 3rd (S).
> > - For the evaluation on "Photo", you train the model on Paintings/Cartoon/Sketch, etc.
> >
> > Is this correct ?
> >
> >
> > > We added an illustration of our method in Figure 1
> >
> > The figure brings more questions than additional information. What is the space represented by the figure ? Which points are training or test points ? How can classifiers also be represented as points ? The caption of the figure is, again, repeating the same blurb ("adapting to a single test sample") without providing additional information/explanations.

---

> > > ### Author Response · Authors · 2021-11-23
> > > **Further response to Reviewer xA1m**
> > >
> > > We thank Reviewer xA1m for the prompt response and engagement. Much appreciated.
> > >
> > > **Intuition**
> > >
> > > The reviewer is quite right that a single sample from the target distribution can't inform much about the whole target distribution. This is also our intuition. It is the reason why we consider each target sample a domain by itself. Correspondingly, we propose that each target sample adjusts the trained classifier in its own way. This avoids the difficulty of using the limited information available to adapt the classifier to the entire target domain distribution. We further clarified this in the Introduction.
> > >
> > > In the variational inference, we use the amortized inference network to take the source features and a single target sample as the input to generate the adapted classifier. To acquire the ability of adaptation with a single sample we rely on a meta-learning-like setting.
> > >
> > > **Meta-learning settings**
> > >
> > > While our method is indeed not a typical meta-learning with bi-level optimization like MAML (Finn et al., 2017, Li et al., 2018), we believe we follow the spirit of meta-learning to achieve our domain generalization ability. In Related Work we mentioned the different semantics for meta-learning used in the domain generalization literature (Du et al., 2020, Zhou et al., 2021, Xiao et al. 2021).
> > >
> > > **Generation of the source and target domains**
> > >
> > > The reviewer is correct.
> > >
> > > **Figure 1**
> > >
> > > We regret that the figure confuses the reviewer. Our intent is to represent the feature space and the learned classifiers. As in the answer for Reviewer n6z3, the classifier consists of vectors, each of which can be seen as a prototype for a category generated by the source features (training points). The black points denote the classifier prototypes. Without adaptation, a target sample (test point) is simply classified by the closest category prototype. With our adaptation, the source samples and the single target sample together define new prototypes of the classifier. We updated the caption of the Figure. Thank you.

---

### Official Review · Reviewer_n6z3 · 2021-10-28

**Correctness:** 3
**Technical Novelty And Significance:** 2
**Empirical Novelty And Significance:** 2
**Recommendation:** 8
**Confidence:** 4

**Main Review:**

# Strengths
1. The idea of artificially creating meta-source and meta-target domain within the source domain is interesting.  This exposes the classifier to the notion of distribution shift and the learnt abilities of the model to handle such distribution shifts are shown to transfer to the unseen target domain.
2. The problem an interesting angle for viewing domain generalization. The key question of "how to expose models to distribution shift" when no target samples are available, is an interesting one.

# Weaknesses
1. It is not clear why a random division of source $\mathcal{S}$ into $\mathcal{S}^\prime$ and $\mathcal{T}^\prime$ will always simulate distribution shift. Does a more sophisticated division (perhaps using clustering or some notion of feature distance) make more sense to create $\mathcal{S}^\prime$ and $\mathcal{T}^\prime$?   The distribution shift between meta-source and meta-target has not been quantified or even visualized.  This limits the modification of this idea to other tasks since _why a random split works_  isn't clear.
2. While the multi-source domain generalization setting has been explored, it is not clear to me whether the same approach will apply to cases where only one source is available (for instance SVHN as source and MNIST/USPS/... as target, as shown by TENT (Wang et al. )).  Domain generalization is largely an unsolved problem in the digits classification task -- which is considered a trivial task in terms of in-domain accuracy (near-100% performance on MNIST).  It is unclear to me how the proposed approach would be able to simulate a meta-source and meta-target if the source is a largely homogenous dataset such as MNIST.
3. It is stated in the introduction that existing test-time adaptation methods such as TENT perform finetuning, while this method does not.  Unless I am mistaken, model parameters are updated for every new test sample even in this method (please correct me if I'm wrong).  So how is it not finetuning?  Sure, additional networks or proxy self-supervision tasks are not used (like in TTT), but there does seem to be adaptation at test time.

# Questions
1. In the PACS experiments (Table 3), the method surpasses prior art with ResNet50, but not with ResNet18. What might be the reason for this?
2. In Figure 2 (and similar figures in the Appendix) I don't understand the underlying space of the visualization.  How is a classifier and a test sample visualized in the same space? Or does the blue symbol indicate the subspace which the classifier predicts as the respective classes? Please provide details of how this visualization is computed.

# Feedback
- On page 1, existing test-time-training methods mentioned (Sun et al, Wang et al) are mentioned, with the statement that "these methods typically rely on batches of samples from the target domain".  This is not entirely true
    - TTT (Sun et al.) have two variants of their method, one which operates on a single sample, and an online method that uses a batch of training samples.
    - There is also a recent method in NLP (Banerjee et al. NAACL 2021 https://arxiv.org/abs/2103.11263 which is largely inspired from TTT) that makes predictions on single test instances.
- Table 1 is useful for understanding how this work differs from prior art.  However TTT (Sun et al.) isn't listed here -- it should be listed as a method that uses single samples $x_t$  and performs fine-tuning/adapting.
- Table 1 could also include a column that states whether extra networks / models are needed.
- The model architecture figure from supplementary material should be moved to the main paper.
- I would also recommend adding an algorithm / pseducode to complement the equations in Section 3. The methods section is more than 2 pages long, and not having a succinct pseudocode hampers the readability of this paper.
- Related to my statement in Strengths #2, a recent paper approaches this problem by proposing a data augmentation method that exposes classifiers to "novel views" of a source image: see Chai et al. CVPR 2021 https://arxiv.org/abs/2104.14551 . However, they target adversarial robustness (and do not have results on DG benchmarks).  It might be interesting to compare their method of test-time data augmentation, with your method of test-time variational inference, in related work.

# Review Summary
The idea is interesting and the paper is well-structured in general.  However there are a few confusing aspects that I have pointed out as well as some inconsistencies with the claims in the paper.  That being said, the empirical results suggest that this method is superior to TTT, TENT and other baselines on benchmarks used in this paper.  I am giving a weak reject at this point, but will be happy to receive clarification from the authors before the final decision.

## update after discussion period
The authors have sufficiently addressed concerns raised by me and other reviewers and new experiments / analyses / comparisons have strengthened the paper.  After these changes, I am increasing my rating to "accept".


**Summary Of The Paper:**

The paper deals with the problem of domain generalization, specifically performing inference on a single test example.  The key idea is to use meta-learning to train the model on the source data -- in the training stage, the source data is divided into meta-source and meta-target, and the model is trained to adapt to the meta-target domain.  This mimics domain shift within the source data.  The method is seamlessly transferred to the target domain, where inference is performed on a single test example, without finetuning model parameters or using additional networks.

**Summary Of The Review:**

interesting idea, but confusing/inconsistent claims; unclear why a random split of source into meta-source and meta-target would create a distribution shift.

---

> ### Author Response · Authors · 2021-11-22
> **Response to Reviewer n6z3**
>
>
> **Divisions of source domains**
>
> We apologize for the unclear description of the data split under the meta-learning setting. In our implementation, the source data is divided into different source domains according to their domain annotations. The random (domain) split of meta-source and meta-target means that we randomly select one of the available source domains as the meta-target domain per iteration. We clarified the division of meta-source and meta-target domains in Section 3.
>
> We did the experiments with the random image split of meta-source and meta-target, as also suggested by Reviewer wnp4. The results in Table 5 in Appendix B show that the random split by images performs worse than the split by domains, indicating the importance of constructing appropriate domain shifts during training.
>
> **Experiments of single-source domain generalization**
>
> As our method learns the adaptation ability by mimicking the domain shift during training, it requires at least two source domains to support the meta-learning scheme. We discussed this imitation of our method in Section 5.
>
> To honor your request, we conducted an experiment that uses SVHN as the single source domain and MNIST as the target domain. We generate source domains by two methods: random image split and clustering of images (K-means). The results in Table 8 show that our method performs slightly better with larger domain shift simulated in the single source domain. To learn better adaptation ability from a single source domain, auxiliary methods are needed to generate larger domain shifts, for example, domain augmentation techniques suggested by Qiao et al. (2020).
>
> | Methods                  | SVHN -> MNIST |
> | ----------------------- | ------------- |
> | Baseline                | 82.05±0.34   |
> | Split by images (ours)  | **82.45**±0.70   |
> | Split by cluster (ours) | **83.01**±0.36   |
>
> **Difference from fine-tuning methods**
>
> Our method is different from fine-tuning in that it does not do backpropagation at test time. The adaptation is achieved by generating the adapted classifiers for each target sample with only one forward pass using the learned amortization inference network. We clarified the difference in Section 2.
>
> **Analysis of the performance with ResNet50**
>
> We explained this in Section 4 (comparisons with state of the art) as: “This is reasonable since we generate the adaptive model using the features of the given target sample. The features extracted by ResNet-50 contain more information, leading to a more adapted model for the given target sample.”
>
> **Details of the visualizations in Figure 2**
>
> We regret our visualization was not clear. In Figure 2, the blue symbols indicate the classifier vectors of different categories, while the red ones indicate the features. The classifier consists of vectors, each of which can be seen as a prototype for a category. The vectors have the same dimension as the test features. Therefore, we can visualize the classifier vectors of different categories and the test features in the same plane. The distance between classifiers and features reflects their similarity. We added these clarifications in Section 4.
>
> **Feedback**
>
> We fixed the wrong statement in the Introduction according to the two references and improve Table 1 as suggested. We also moved the architecture diagram to the main paper. For clarity and simplicity, we made some slight adjustments to the figure. Due to lack of space, we inserted the algorithm in the Appendix. The comparison with Chai et al. (2021) was also added in Related Work. Thank you.

---

> > ### Comment · Reviewer_n6z3 · 2021-11-24
> > **Not convinced about ResNet18 vs ResNet50.  Comparison when using k-means vs domain labels needed for multi-source DG.**
> >
> > - table 1 looks more complete now -- thanks for the changes
> > - ` In our implementation, the source data is divided into different source domains according to their domain annotations` -- thanks for the clarification -- crucial for my evaluation of the paper
> > - `We did the experiments with the random image split of meta-source and meta-target` -- yes intuitively this should perform worse than if you have domain labels.
> > - `Difference from fine-tuning methods .. no backprop .. only 1 forward pass` -- thanks, this clarification is definitely needed.
> >
> > - Resnet18 vs Resnet50 -- `“This is reasonable since we generate the adaptive model using the features of the given target sample. The features extracted by ResNet-50 contain more information, leading to a more adapted model for the given target sample.”`
> >     - this may be the case, but this sounds like intuition, not sure if there is enough empirical evidence for this statement
> >     - do you observe better and better performance gap between basic ERM and your method if you use larger and larger models?
> >
> > Now lets move back to the domain split method -- as you said, you use additional "domain annotations" to create splits. This makes sense if you have multiple domains available.  Then in SSDG, you showed that you can split using k-means clustering and that is better than ERM -- this is a good finding.
> >
> > I wonder if the same idea can also be applied for MSDG? Say you have n source domains, but don't have labels -- then would some sort of clustering give you a good split and similar performance improvements? i.e. how much of the performance improvement can be attributed to the availability of domain labels?
> > Perhaps comparing ERM vs This Paper (k-means clustering) vs This Paper (using domain labels)  would help for completeness.  The method can be more impactful if simple domain splitting methods like clustering can also give performance improvements in MSDG (often in realistic settings, domain labels are not available)/
> >
> > I'm still on the borderline -- if you could clarify these points -- that would be great.  I understand that there may not be enough time, so any preliminary experiments / insights / etc. will also help me make my final recommendation.

---

### Official Review · Reviewer_dNDg · 2021-10-30

**Correctness:** 3
**Technical Novelty And Significance:** 3
**Empirical Novelty And Significance:** 2
**Recommendation:** 8
**Confidence:** 4

**Main Review:**

Pros.

1. The proposed method is more flexible than the previous generalization- and adaptation-based methods. Compared to generalization-based methods, the model can better adapt to target domains via conditional generated parameters. Compared to previous adaptation-based methods, the proposed method does not rely on batches of samples from the target domain with extra fine-tuning operations or extra networks.

2. The paper systematically summarizes previous related works including domain adaptation/generalization, domain meta-learning, and test-time adaptation. Table 1 clearly shows the difference between this work and previous works in terms of training and test-time settings.

Cons.

1. The proposed method requires multiple source domains during training to support the meta-learning scheme. However, previous test-time adaptation models do not have such limitations and can be trained on one single domain. How to guarantee fairness in comparison to previous test-time adaptation methods? What is the minimal number of source domains required for training?

2. During training, the paper proposed to approximate the model function with the accessible source data by the variational inference. Is there any underlying assumption about the distribution similarity between the source and target domain? Can the "adaptivity gap" still be minimized when the target domain is significantly different from the source?

3. The paper proposed to construct an adapted classifier during inference. As indicated in the main paper, "both the meta-prior distribution and the variational posterior distribution of the classifier are generated by amortized inference using the amortization technique", but why the amortized inference is applied?  More elaboration is required to justify the motivation.


**Summary Of The Paper:**

This paper studied the problem of single test sample generalization. Its goal is to adapt a pre-trained model to unseen target domains without extra fine-tuning. The paper formulated the single test sample generalization problem as a variational inference problem and proposed a meta-learning framework. To bypass extra fine-tuning, a single test sample was used as a conditional to generate model parameters. Experiments and ablation studies on common-used benchmarks demonstrate the effectiveness of the proposed method.

**Summary Of The Review:**

This paper addressed "single test sample generalization", which is an interesting and important problem in the field of domain generalization. The method is technically sound and novel. Although I have some concerns regards the method and experiments as indicated in the main review, I overall remain positive towards this paper. I hope the authors can address my concerns in the rebuttal.

After rebuttal: I have read the response and comments of other reviewers. All of my concerns have been addressed in the response. I recommend acceptance for this paper. I highly suggest that the authors should widely discuss the remaining weaknesses raised by the reviewers in the final version.

---

> ### Author Response · Authors · 2021-11-22
> **Response to Reviewer dNDg**
>
> **Fair comparisons with test-time adaptation methods**
>
> The reviewer is correct that previous test-time adaptation models can adapt to unseen target domains with models trained on a single source domain, which is one of our limitations. In our method, we need at least two source domains during training (one serves as the meta-source domain while the other serves as the meta-target).
>
> Under the multi-source domain generalization setting, we train the base model using the same amount of train and validation image data as in our method (be it we split the images in our method over multiple domains by their domain annotations). Then the trained base model for Tent is adapted by the target samples, as in their original paper. The other settings are the same for the two methods.  We believe this to be fair and we added the detailed training settings of Tent in Section 4.
>
> **Similarity assumption**
>
> Our method does not necessarily rely on the underlying assumption about the distribution similarity since the method incorporates the single target sample in the variational posterior and learns the ability to adapt to single target samples through the meta-learning scheme. To learn the ability of adaptation to any target domain during training, our method needs at least two source domains that have large enough domain shifts. This limitation has been discussed in Section 5.
>
> **Motivation of amortization inference**
>
> Our method needs to incorporate the feature of the target sample into the construction of the adapted variational posterior distribution. The amortization technique provides a natural way to generate model parameters by feature representations. Moreover, the amortization networks can be trained and evaluated together with the model, without introducing extra training steps and fine-tuning operations to achieve the adaptation ability. We clarified the motivation in Section 3. Thank you.

---

> > ### Comment · Reviewer_dNDg · 2021-11-28
> > **Thanks for your response**
> >
> > I thank the authors for their response, and all of my concerns are addressed by the rebuttal. After reading the other reviews and the corresponding responses, I share the same impression with Reviewer wnp4 that the remaining issues lie primarily in the clustering-based split construction experiments for MSDG and low performance when ResNet-18 is used as the backbone. However, I think the former one is good to have but might not play a vital role in judging the contribution of this paper since this setup (i.e., having access to multiple source domains without knowing their domain ids) is rare in real-world applications. At the current point, given not only the novelty but the interesting setting this paper addressed, I am willing to increase my score and recommend acceptance if the concern about ResNet-18 results can be properly addressed.

---

### Official Review · Reviewer_6n8V · 2021-11-03

**Correctness:** 4
**Technical Novelty And Significance:** 3
**Empirical Novelty And Significance:** 2
**Recommendation:** 5
**Confidence:** 3

**Main Review:**

Strength:
* The paper proposes to use meta-learning scheme to do one-shot learning. The idea is heuristic to the research community
* The paper provides several empirical study and analysis to demonstrate the performance of the proposed approach
* the proposed method can benefit real application scenario of extreme few shot cases


Weakness:
* It is unclear how training on each instance of the source domain would enable a large enough domain shift to benefit a larger domain shift encountered during test time.
* even though most experiments in the paper are considering multi-source domain adaptation. It is unclear why not testing on the single-source setting, since training the proposed method does not require multiple source domains
* It seems that the proposed method underperforms the method proposed by Zhou et al. (2020a)  in all of the Resnet-18 experiments. and For some reason, for all Resnet-50 Experiments only for this baseline method is not conducted
* it would be helpful to show how an ablation study on how exactly the proposed meta-learning scheme improves performance vs non-meta-learning based one-shoot learning, such as [1, 2],

[1] Luo, Yawei and Liu, Ping and Guan, Tao and Yu, Junqing and Yang, Yi. Adversarial Style Mining for One-Shot Unsupervised Domain Adaptation. Advances in Neural Information Processing Systems, 2020.

[2] Dong, Nanqing and Eric P. Xing. Domain adaption in one-shot learning. Joint European Conference on Machine Learning and Knowledge Discovery in Databases, 2018. 573--588.

**Summary Of The Paper:**

The authors propose to learn to generalize across different target domains with single samples by using a meta-learning paradigm. Specifically, during training, the course domains are divided into several meta-source domains and meta-target domains to explore the adaptivity of the model by incorporating information of the meta-target sampling when learning the model parameters.

**Summary Of The Review:**

As the discussed above, due to weak performance, and lack of more empirical demonstration of the proposed method. I recommend marginal reject on this paper. I would like to adjust my rating after discussing with other reviewers and AC

---

> ### Author Response · Authors · 2021-11-22
> **Response to Reviewer 6n8V**
>
> Before answering the questions, we would like to clarify the difference between our setting and one-shot domain adaptation. In one-shot domain adaptation, the methods require an (unlabeled) target sample to be available during training. By contrast, our method assumes not a single target sample is accessible during training, see also Table 1. Instead, we construct our model under a meta-learning framework to learn the ability of adaptation to unseen target samples across (at least two) source domains. At test time, no more learning is needed and we generate an adapted model for each individual target sample by only a forward pass, without any extra fine-tuning operation.
> We further clarified the differences in Related Work.
>
> **How our method benefits the domain shifts encountered during test-time**
>
> Rather than simply training on each instance of the source domain, we train the model to handle domain shift under a meta-learning framework. During training, our model learns the ability to adapt the meta-source model to each meta-target instance across different domain shifts within multiple source domains. The learned ability is exploited to handle domain shifts at test time by adapting the source model to each (unseen) target instance. We clarified this in Section 3.
>
> **Single-source setting**
>
> We would like to clarify our meta-learning method requires at least two source domains during training to learn the ability to address domain shift. To handle the single-source setting, auxiliary methods are required to simulate the domain shift during training, e.g., domain augmentation. We discussed this limitation in Section 5 “Conclusion and Future Work”.
>
> To honor your request, we conducted single-source domain generalization experiments by generating multiple source domains with two methods: random split images and clustering of images (K-means). The experiments use SVHN as the source domain and MNIST as the target. The results are shown in Table 8 in Appendix B. Our method performs slightly better with larger domain shift simulated in the single source domain. To achieve better adaptation, further auxiliary methods like domain augmentation are required to construct domain shifts during training.
>
> | Methods                  | SVHN -> MNIST |
> | ----------------------- | ------------- |
> | Baseline                | 82.05±0.34   |
> | Split by images (ours)  | **82.45**±0.70   |
> | Split by cluster (ours)  | **83.01**±0.36   |
>
> **Comparisons with Zhou et al. (2020a)**
>
> The reviewer is correct that we underperform the method of Zhou et al (2020a) for ''art'' and ''real world'' in Office-Home and ''art painting'' and ''photo'' in PACS using a ResNet-18 backbone. Zhou et al. (2020a) do not provide ResNet-50 results, but our reimplementation indicates that our method performs better based on ResNet-50 as the extracted features contain more information, leading to better-adapted models. See also our response to a similar question by Reviewer n6z3. We added the comparisons in Section 4.
>
> | Methods (based on ResNet-50) | photo       | art         | cartoon     | sketch      | mean        |
> | ----------------- | ----------- | ----------- | ----------- | ----------- | ----------- |
> | Zhou et al. 2020a | 97.55       | 85.21       | 80.33       | 76.53       | 84.9        |
> | Ours              | **97.88**±0.15  | **88.09**±0.26  | **83.83**±0.68  | **80.21**±0.66  | **87.51**±0.22  |
>
> |  Methods (based on ResNet-50) | art         | clipart     | product     | real world  | mean        |
> | ----------------- | ----------- | ----------- | ----------- | ----------- | ----------- |
> | Zhou et al. 2020a | 63.62       | 51.48       | 76.57       | 78.95       | 67.66       |
> | Ours              | **67.21**±0.59  | **57.97**±0.37  | **78.61**±0.59  | **80.47**±0.16  | **71.07**±0.31  |
>
> **Ablation study on the meta-learning scheme**
>
> One-shot learning methods are not applicable to our more strict domain generalization setting, as these methods require an unlabeled target sample to be available during training. The discussion about the difference from these methods (Dong & Xing, 2018 and Luo et al., 2020) is added in Section 2.
>
> To still honor your request, we implemented a non-meta-learning version of our method. The results in Table 6 in Appendix B show the benefits of training by meta-learning. Thank you.
>
> | Settings     | photo       | art         | cartoon     | sketch      | mean |
> | ----------- | ----------- | ----------- | ----------- | ----------- | ----------- |
> | Non-meta    | 94.61±0.65 | 79.97±0.43 | 78.45±0.29 | 75.83±0.79 | 82.21±0.36 |
> | Meta (ours) | **95.87**±0.24  | **82.02**±0.36  | **79.73**±0.49  | **78.96**±0.67  | **84.15**±0.21 |

---

### Official Review · Reviewer_wnp4 · 2021-11-03

**Correctness:** 4
**Technical Novelty And Significance:** 3
**Empirical Novelty And Significance:** 3
**Recommendation:** 8
**Confidence:** 4

**Main Review:**

I will now highlight the strengths and weaknesses of the paper.

**Strengths**

1. The paper is generally well-written and easy to follow. With the exception of a few minor points raised under weaknesses, the authors generally do a good job of situating the problem setup and motivating the proposed approach. I particularly appreciate how the authors build up from the most rudimentary version of the proposed approach to adding more relevant terms to the objective that might further improve out-of-domain performance. Furthermore, in this space of test-time adaptation, when compared to prior work which either rely on single or multiple passes over the entirety of target data, I think the paper is trying to address a relatively realistic and timely problem setting that the community might find useful.

2. Unlike prior test-time adaptation approaches, I think a big positive point in support of the proposed approach is that it doesn’t require — (1) fine-tuning on target data, (2) more than a few target samples, and still ends up either being competitive or outperforming the same. Obtaining an adapted model on target data based on a single or few source samples with a single forward pass offers significant benefits in terms of data and computational constraints.

3. Out-of-domain generalization results obtained via the proposed approach seem to demonstrate the utility of the same (although marginally in some cases). Further ablations on rotated-MNIST highlight the stability and utility of the same when compared to TENT. Additionally, I like that the authors' highlight failure cases of the proposed approach accompanied with hypotheses surrounding things that fail on particular samples.

**Weaknesses**

1. If I understand correctly, one of the crucial bits the proposed approach relies on during training is the availability of domain labels on source data — source instances belong to different domains because the domain labels say so, which in turn influences how the meta-source and meta-target splits are constructed (also noted under conclusions in the submission). While this is a perfectly fine assumption, the paper would benefit if the authors included an experiment validating the extent to which the proposed approach relies on this. Essentially, a setting where the source domain labels are randomized — as in, (say) if there are 3 source domains, data at training time is grouped randomly into 3 groups instead of by domain and meta-source & meta-target splits are constructed based on these three groups. Considering such a setting would highlight how important mimicking the domain shift within a source is important. If the out-of-domain obtained results are worse, then there is additional validation that constructing appropriate meta-source & meta-target shifts matters. If not, then the utility of the proposed approach lies somewhere else. I would encourage the authors to include such an experiment.

2. [Minor Points] Including the model architecture diagram from the appendix to the main paper would make it easier for the reader to follow the supporting description in Section 3  (page 5) and under implementation details. While I understand the need to include the target sample in the variational posterior (as considered in equation 3), I don’t completely follow the supporting justification (on page 4, the paragraph above “single-test sample generalization”) in terms of the adaptivity gap. Adaptivity gap (as discussed in Dubey et al. 2021) is a general statement about DG settings. I’m not entirely certain what extra insights it provides in the supporting justification. When finally evaluating on target data, is an adapted model obtained for every target sample, or is it the case that it is obtained from a few samples. If it’s the latter, how are these target samples chosen?

**Summary Of The Paper:**

The paper proposes an approach to ensure a model trained on a set of source domains generalizes well to an unseen target domain based on a single unlabeled target sample. Unlike the standard domain generalization (DG) setting, where there is no scope for adaptation in the target domain, and domain adaptation (DA), where the model has access to source and unlabeled target data during training, the paper falls somewhere in between where no access to target data is assumed during training and a single unlabeled target sample is allowed for quick adaptation. In principle, the proposed setting is somewhat similar to source-free domain adaptation with the distinction that only one sample (and not the entire target dataset) and one quick adaptation pass on the data is allowed. The authors leverage a meta-learning paradigm to mimic domain shift by defining meta-source and meta-target domains within the source domain vocabulary (has also been considered in prior DG settings). The approach adopted by the authors relies on modeling a conditional distribution over the model parameters given a meta-target domain sample and source domain representative features. The authors set this up within a variational inference framework that allows them to explicitly parameterize this conditional distribution to infer model parameters in a target domain. The authors further explore the utility of different instantiations and alternative formulations of their objective and show that their proposed version has the best performance. When compared with prior DG and test-time adaptive methods, the proposed approach leads to competitive or improved performance. Further ablations demonstrate the utility of the proposed approach over other test-time adaptive approaches in terms of number of target samples available for unsupervised adaptation.

**Summary Of The Review:**

The points highlighted under strengths and weaknesses form the basis of my rating. I am generally supportive of the paper and think that it addresses the test-time adaptation problem from a relatively constrained yet realistic standpoint. The paper is well-written and generally easy to follow. The most positive bit in support of the paper in my opinion is the lack of reliance on the entirety of target data or multiple passes on the same to adapt a model. Regarding weaknesses, since the paper relies heavily on the meta-source and meta-target split construction, I think it’s important to address the extent to which this reliance is true (in accordance with the suggested experiments in 1). The minor points in 2 are addressable, I think.

**Thoughts post author responses**

As stated in my follow-up reply to the authors, I think my primary concerns regarding domain-split construction were sufficiently addressed by the authors. Similar concerns were also shared by other reviewers and the authors provided sufficient experimental evidence to address the same. Additionally, concerns surrounding results with different backbones have also been sufficiently addressed (in my opinion) by the new experimental results. Given these and the fact that the paper makes an interesting contribution and studies a timely experimental setting, I continue to recommend acceptance of the paper (and would like to stick to my original rating).

---

> ### Author Response · Authors · 2021-11-22
> **Response to Reviewer wnp4**
>
> **Experiments of domain division by randomly splitting images**
>
> The reviewer is correct that our method relies on the domain labels. We conducted the suggested experiments with a random image split over source domains on PACS with a ResNet-18 backbone. Results in the new Table 5 in Appendix B show that by randomly splitting images from source data into three domains, the performance on each domain drops, especially on “art_painting” and “sketch”, confirming the importance of constructing appropriate domain shift within source domains during meta-learning.
>
> | Division of source domains  | photo | art-painting | cartoon | sketch  | mean|
> | --------------------------- | ----------- | ------------ | ----------- | ----------- | ----------- |
> | Random split by images | **95.75**±0.24 | 78.30±0.33  | 78.52±0.81 | 75.69±0.64 | 82.07±0.11 |
> | Split by domain annotations | **95.87**±0.24  | **82.02**±0.36  | **79.73**±0.49  | **78.96**±0.67  | **84.15**±0.21  |
>
> **Minor points**
>
> We moved the model architecture diagram to Section 3 in the main paper. For clarity and simplicity, we made some slight modifications to the diagram as well.
>
> The insight provided by the “adaptivity gap” is that the adaptivity gap always exists in domain generalization, as the target data is never seen during training. Thus, a model needs to acquire the ability to effectively use the target information at test time to reduce the adaptivity gap. Therefore, we introduce the meta-target sample (from source domain data) into the distribution during training. We added this discussion in Section 3.
>
> In the final evaluation, we obtain an adapted model for each sample in the unseen target domain, so the method does not need to choose target samples. Thank you.

---

> > ### Comment · Reviewer_wnp4 · 2021-11-25
> > **Thanks for the response**
> >
> > Apologies for the delay and thanks to the authors for responding to my concerns. I read the other reviews and the corresponding responses to the same. Highlighting below which concerns (mine or otherwise) were sufficiently addressed.
> >
> > Starting with mine, I essentially had one primary concern centered around the degree to which the proposed approach relies on domain labels. I suggested an experiment where the meta-target (and source) splits were constructed by random domain label assignments. The authors responded with results for the same in the multi-source DG setting on PACS. The obtained results imply that overall, proper split construction during training does indeed result in improved out-of-domain performance (the difference being roughly ~1.8 %). Additionally, given the results from the single source DG experiments (using random split assignments and clustering) presented in the responses to reviewers 6n8v and n6z3, the conclusion seems to be that appropriate domain split construction helps significantly when the source data has diversity in terms of underlying domains (either explicitly accessible via domain labels or implicitly via some additional step). This seems like an important observation and I would strongly encourage the authors to add a discussion surrounding the same in the draft since it might serve as a useful point of reference for future approaches attempting to build on top of the one proposed in this paper. I think my concern surrounding this point has been sufficiently addressed.
> >
> > My other concerns were minor clarification concerns and edits. I think the authors have sufficiently addressed those.
> >
> > I will now discuss the concerns raised by other reviewers. I think the concerns surrounding comparisons with test-time adaptation methods (within the multi-source DG setting) and assumptions about distribution similarity have been sufficiently addressed. Regarding the point about performing the clustering-based split construction experiments during training for the multi-source setting (latest response from reviewer n6z3), I think not including this experiment isn’t necessarily detrimental to the contributions of the paper. The MSDG experimental results provided in the response to my concerns — were annotation-based split construction is compared with a randomized version — do imply that utilizing domain label annotations help over using randomized ones. In terms of cluster-purity (where purity is calculated with the ground-truth domain labels), I would expect out-of-domain results using label assignments based on clustering to lie somewhere in between the randomized and the domain label versions. However, I do think this experiment is useful — in the sense that it might highlight the extent of domain label purity required for the proposed approach to work (and can potentially improve the proposed approach if clustering works well in terms of DG results). I would encourage the authors to include this experiment. The only weak bit right now seems to be the response to the concern surrounding the results using ResNet-18 and ResNet-50. I agree with reviewer n6z3 that the provided response seems insufficient and weak.
> >
> > Overall, I think the paper makes an interesting contribution and studies a timely experimental setting. I think my concerns have been sufficiently addressed. As stated earlier, the only weak bit seems to be the response to the concern about resnet results. Given this, I would like to stick to my original rating of the paper.

---

> > > ### Comment · Reviewer_wnp4 · 2021-11-30
> > > **Thoughts post new results**
> > >
> > > Thanks for sharing the new experimental results with (1) different backbones and (2) meta-source / target split construction methods. Given that these results continue to solidify (and address the feedback to) the responses to the points raised in the discussion (including me and other reviewers), I continue to recommend acceptance of this paper.
> > >
> > > As stated by the authors, I think including these results and a supporting discussion will definitely improve the paper.

---

### Author Response · Authors · 2021-11-22
**Summary of changes**

We thank all Reviewers for their insightful reviews, sharp comments, and supportive suggestions. Here, we provide a summary of the updates made in the new version per section, as suggested by the reviewers. We provide an individual response to each reviewer separately.

The following updates have been incorporated in the  **main manuscript**:

* In Section 1, we fixed the wrong statements and improved Table 1. We also provided the intuition and an illustration figure of our method.
* In Section 2, we added the discussion on the difference of our method from one-shot domain adaptation and fine-tuning methods. We also discuss the meta-learning scheme utilized in our method.
* In Section 3, we further clarified the meta-learning setting in our method, e.g., the division of source domains, the training and inference stage. We added the motivation for applying amortization inference. We also moved the architecture diagram from Appendix to Section 3.
* In Section 4, we added more details about the visualizations. We also added the comparison with Zhou et al. (2020a) based on ResNet-50 in Tables 3 and 4. We provided detailed training settings of Tent for fair comparisons and added the analyses on why our method performs well.

The following updates have been inserted in the **Appendix**.
* In Appendix B, we added experiments on different domain divisions in Table 5 and an ablation study of the meta-learning scheme in our method in Table 6. We also provide single source domain generalization experiments in Table 8.
* We added a pseudo-code algorithm in Appendix C.
* We moved the implementation details and the comparison on rotated MNIST and Fashion-MNIST to Appendix D.

---

### Author Response · Authors · 2021-11-25
**Further response to Reviewers wnp4, n6z3, and dNDg**

We thank Reviewers wnp4, n6z3, and dNDg for their response, engagement, and encouragement. Appropriate domain split construction is indeed important for our method and we are excited by the realization it can be relaxed, for example by clustering. We will add this discussion to the paper.

We are currently performing the requested experiments for the clustering-based domain split in the multi-source setting and the comparisons with the ERM method for different backbones. We will post the results once the experiments are finished, which we expect will happen before the closing of the discussion stage on Nov 29th. Thank you.

---

> ### Author Response · Authors · 2021-11-27
> **Clustering-based experiments in the multi-source setting**
>
>
> We finished the clustering-based domain split on the multi-source setting of PACS, based on a ResNet-18 backbone. We use K-means to generate three clusters of images from three source domains. We provide the results together with the results of the random image split and the domain annotation-based split, which we provided in Appendix B in the updated paper.
>
> | Division of source domains  | photo | art-painting | cartoon | sketch  | mean|
> | --------------------------- | ----------- | ------------ | ----------- | ----------- | ----------- |
> | ERM baseline | 94.71±0.24 | 77.98±0.31  | 77.78±0.39 | 75.34±0.66 | 81.45±0.21 |
> | Random split by images | **95.75**±0.24 | 78.30±0.33  | 78.52±0.81 | 75.69±0.64 | 82.07±0.11 |
> | Split by cluster of images | **95.67**±0.39 | 80.05±0.56  | **79.39**±0.60 | 77.12±1.00 | 83.06±0.39 |
> | Split by domain annotations | **95.87**±0.24  | **82.02**±0.36  | **79.73**±0.49  | **78.96**±0.67  | **84.15**±0.21  |
>
> As hinted by Reviewers n6z3 and wnp4, the cluster-based domain split lies between the split by images and the split by domain annotations on three of the four domains, demonstrating the importance of constructing appropriate domain shift during training. Importantly, even with a simple unsupervised domain split method, like clustering, we outperform a baseline ERM model considerably. We will add these experiments, findings, and discussions in Section 4 in the main paper. Thank you.
>
> The experiments with different backbones are still ongoing, we will provide the results once they are finished.

---

> ### Author Response · Authors · 2021-11-28
> **Experiments with different backbones**
>
> In response to the request by Reviewers n6z3, wnp4, and dNDg, we compare our proposal with a baseline ERM method on PACS by varying the backbone model size. The ERM baseline is trained and evaluated using the same settings and hyperparameters as our method, which is shown in the implementation details in Appendix D. All experiments are averaged over five runs. The results are as follows:
>
> |           | ERM baseline |            |             |            |            | Ours       |            |            |            |            |          |
> | --------- | ------------ | ---------- | ----------- | ---------- | ---------- | ---------- | ---------- | ---------- | ---------- | ---------- | -------- |
> | backbone  | Photo        | Art        | Cartoon     | Sketch     | Mean       | Photo      | Art        | Cartoon    | Sketch     | Mean       | Mean gap |
> | AlexNet   | 88.47±0.32  | 66.64±0.37 | 69.25±0.43 | 60.55±0.42 | 71.23±0.38 | 89.70±0.35 | 67.14±0.30 | 71.71±0.56 | 61.45±0.47 | 72.50±0.28 | 1.27     |
> | ResNet-18 | 94.71±0.24  | 77.98±0.31 | 77.78±0.39  | 75.34±0.66 | 81.45±0.21 | 95.87±0.24 | 82.02±0.36 | 79.73±0.49 | 78.96±0.67 | 84.15±0.21 | 2.70      |
> | ResNet-34 | 94.73±0.25  | 80.36±0.37 | 78.54±0.49 | 76.70±0.69 | 82.58±0.36 | 96.35±0.25 | 84.73±0.21 | 82.82±0.63 | 79.72±0.83 | 85.91±0.38 | 3.33     |
> | ResNet-50 | 95.65±0.18  | 82.34±0.46 | 78.79±0.64 | 79.55±0.65 | 84.09±0.13 | 97.88±0.15 | 88.09±0.26 | 83.83±0.68 | 80.21±0.66 | 87.51±0.22 | 3.42     |
>
> With larger and larger backbone models, our method achieves a better and better performance gap compared to the ERM baseline. We explain this by our use of an amortized inference network that generates the target sample classifier by the source domain features and the target sample feature. Implying an increase in backbone capacity has a direct effect on our classifier capacity, enabling better adaptation to the target sample. We will add the experiments and discussion in the paper.
>
> We hope to have mended the remaining concerns and thank all reviewers for the insights, the encouragement, and the help to improve the paper.

---

> > ### Comment · Reviewer_n6z3 · 2021-11-30
> > **Final response from reviewer n6z3**
> >
> > Hi authors,
> >
> > Thanks for promptly responding to each concern from the reviewers -- appreciate the additional experiments that you have conducted to answer our questions.  The comparison with different model sizes (Alex/ResNet18/34/50) is also useful.
> > These results have definitely improved my understanding of the method and addressed my concerns.
> > Overall, I am going to recommend *acceptance*.
> >
> > After reading the latest responses from other reviewers, I support the statement from reviewer wnp4 and encourage you to consider it in the final draft:
> > ```
> > Additionally, given the results from the single source DG experiments (using random split assignments and clustering) presented in the responses to reviewers 6n8v and n6z3, the conclusion seems to be that appropriate domain split construction helps significantly when the source data has diversity in terms of underlying domains (either explicitly accessible via domain labels or implicitly via some additional step). This seems like an important observation and I would strongly encourage the authors to add a discussion surrounding the same in the draft since it might serve as a useful point of reference for future approaches attempting to build on top of the one proposed in this paper.
> > ```
> > wnp4 has also provided a good summary of all reviews (thank you!) and since I agree with it, I will not repeat those points -- glad the authors found our comments/discussion useful.
> >
> > Best of luck.

---

### Decision · Program_Chairs · 2022-01-20

**Decision:**

Accept (Poster)

**Comment:**

This paper proposes a new method for domain generalization by adopting a single test example. Authors formulate the problem using a variational bayesian framework which ends up in an adaptation technique requiring a single feed-forward computation. The provided empirical results indicate that the proposed method has comparable performance to techniques which require more data.

Reviewers all acknowledge the novelty and significance of this work. The paper is well-written and the related work is adequately discussed. Moreover, the proposed method is computationally efficient and empirical results provide strong evidence in its favor. While I am recommending acceptance, I tend to agree with reviewer xA1m about the main weaknesses of this work and I recommend authors to improve them for the final version:

- Lack of proper discussion or intuition about under what conditions the proposed method works well. This may be using theoretical analysis, using toy examples, trying to break the method, motivate using prior work or just simply providing intuitive arguments. Also, as reviewers pointed, Figure 1 is currently very confusing.
- Lack of analysis or ablation study allows a better understanding of the proposed method